# A previously unrecognized superfamily of macro-conotoxins includes an inhibitor of the sensory neuron calcium channel Cav2.3

Celeste M. Hackney[1], Paula Flórez Salcedo[2], Emilie Mueller[3], Thomas Lund Koch[4,5], Lau D. Kjelgaard[1], Maren Watkins[6], Linda G. Zachariassen[7], Pernille Sønderby Tuelung[7], Jeffrey R. McArthur[8], David J. Adams[8], Anders S. Kristensen[7], Baldomero Olivera[6], Rocio K. Finol-Urdaneta[8,9], Helena Safavi-Hemami[4,5,6], Jens Preben Morth[3], Lars Ellgaard[1] *

1 Department of Biology, Linderstrøm-Lang Centre for Protein Science, University of Copenhagen, Copenhagen, Denmark, 2 Department of Neurobiology and Anatomy, University of Utah, Salt Lake City, Utah, United States of America, 3 Enzyme and Protein Chemistry, Section for Protein Chemistry and Enzyme Technology, Department of Biotechnology and Biomedicine, Technical University of Denmark, Kgs. Lyngby, Denmark, 4 Department of Biochemistry, University of Utah, Salt Lake City, Utah, United States of America, 5 Department of Biomedical Sciences, University of Copenhagen, Copenhagen, Denmark, 6 School of Biological Sciences, University of Utah, Salt Lake City, Utah, United States of America, 7 Department of Drug Design & Pharmacology, University of Copenhagen, Copenhagen, Denmark, 8 Illawarra Health and Medical Research Institute (IHMRI), Faculty of Science, Medicine and Health, University of Wollongong, Wollongong, Australia, 9 Electrophysiology Facility for Cell Phenotyping and Drug Discovery, Wollongong, Australia

* lellgaard@bio.ku.dk

**Data Availability Statement:** All relevant data, unless listed below, are within the paper and its supporting information files. The atomic

## Abstract

Animal venom peptides represent valuable compounds for biomedical exploration. The venoms of marine cone snails constitute a particularly rich source of peptide toxins, known as conotoxins. Here, we identify the sequence of an unusually large conotoxin, Mu8.1, which defines a new class of conotoxins evolutionarily related to the well-known con-ikot-ikots and 2 additional conotoxin classes not previously described. The crystal structure of recombinant Mu8.1 displays a saposin-like fold and shows structural similarity with con-ikot-ikot. Functional studies demonstrate that Mu8.1 curtails calcium influx in defined classes of murine somatosensory dorsal root ganglion (DRG) neurons. When tested on a variety of recombinantly expressed voltage-gated ion channels, Mu8.1 displayed the highest potency against the R-type (Cav2.3) calcium channel. $Ca^{2+}$ signals from Mu8.1-sensitive DRG neurons were also inhibited by SNX-482, a known spider peptide modulator of Cav2.3 and voltage-gated $K^+$ (Kv4) channels. Our findings highlight the potential of Mu8.1 as a molecular tool to identify and study neuronal subclasses expressing Cav2.3. Importantly, this multidisciplinary study showcases the potential of uncovering novel structures and bioactivities within the largely unexplored group of macro-conotoxins.

## Introduction

Animal venom peptides and proteins are employed for the incapacitation of prey or the defense against predators and competitors [1]. Venom components function by binding with

coordinates the structure-factor amplitudes for Mu8.1_38 and Mu8.1_59 are available with the Protein Data Bank under accession numbers 7PX1 and 7PX2, respectively. Nucleotide sequence data for Mu8.1 and Mu8.1ii have been submitted to GenBank with accession numbers ON755370 and ON755371 respectively.

**Funding:** This work was supported by the Independent Research Fund Denmark, Technology and Production Sciences grant (#7017-00288 to LE). Research conducted at MAX IV, a Swedish national user facility, is supported by the Swedish Research Council under contract 2018-07152, the Swedish Governmental Agency for Innovation Systems under contract 2018-04969, and Formas under contract 2019-02496. CPHSAXS is funded by the Novo Nordisk Foundation (grant no. NNF19OC0055857). Part of this work was funded by a National Institutes of Health grant (NIGMS R01GM144719 to BO and HS-H). HS-H acknowledges a research grant from Villum Fonden (19063 to HS-H). Electrophysiological characterization was performed with support from Rebecca Cooper Foundation for Medical Research (PG2019396 to JRM). JRM and RKF-U were supported by grant funding from the National Health & Medical Research Council awarded to Prof. D.J. Adams (NHMRC Program Grant APP1072113 to DJA). The funders had no role in study design, data collection and analysis, decision to publish, or preparation of the manuscript.

**Competing interests:** The authors have declared that no competing interests exist.

**Abbreviations:** AEX, anion exchange; AITC, allyl isothiocyanate derived from mustard oil; CNGB, China National Genebank; APC, automated patch clamp; CD, circular dichroism spectroscopy; CGRP, calcitonin gene-related peptide; CLANS, CLuster ANalysis of Sequences; C-LTMR, C-low threshold mechanoreceptor; CNS, central nervous system; csPDI, conotoxin-specific PDI; CV, column volume; DDBJ, DNA Databank of Japan; DMEM, Dulbecco's Modified Eagle's Medium; DRG, dorsal root ganglia; DTT, dithiothreitol; EG, ethylene glycol; ER, endoplasmic reticulum; FDA, United States Food and Drug Administration; FOM, figure of merit; GFP, green fluorescent protein; GPCR, G protein–coupled receptor; HEK293T, human embryonic kidney cells; hPDI, human PDI; IACUC, Institutional Animal Care and Use Committee; IB4, Alexa Fluor 647 Azolectin B4; IPTG, isopropyl ß-D-1-thiogalactopyranoside; LB, lysogeny broth; MALDI-TOF, matrix-assisted laser desorption–ionization time of flight; MPC, manual patch clamp; NCBI, National Center for Biotechnology Information; ORF, open reading frame; PDB,

high affinity and selectivity to their molecular targets. These are often specific membrane-bound proteins that control vital cellular signaling pathways and include ligand and voltage-gated ion channels, G protein–coupled receptors (GPCRs), tyrosine kinase receptors, and transporters [2]. Because of the high similarity between venom peptide targets in the prey and their orthologs in mammals, as well as the conservation of signaling pathways, venom components often show activity in mammalian systems. Animal toxins are therefore interesting for the development of molecular probes and biological tools as well as potential drug leads. Currently, 8 venom-derived drugs have been approved by the United States Food and Drug Administration (FDA) for human use, and approximately 30 other venom-derived peptides are in clinical and preclinical trials to treat a variety of diseases, such as diabetes, hypertension, chronic pain, thrombosis, cancer, and multiple sclerosis [3,4].

The venom produced by predatory marine cone snails is particularly rich in peptide toxins (known as conotoxins or conopeptides). Each of the approximately 1,000 extant cone snail species expresses a unique set of several hundred conotoxins [5], resulting in an estimated diversity of more than 200,000 conotoxins. Conotoxins often display exquisite specificity for their targets and are consequently used widely as pharmacological tools for research purposes. Moreover, the ω-MVIIA conotoxin, which inhibits the Cav2.2 calcium channel [6], is an FDA-approved drug (commercial name Prialt) for the treatment of severe chronic pain [7,8].

Conotoxins are used extensively to investigate ion channel function (and dysfunction) as illustrated by the κM-conotoxin RIIIJ from *Conus radiatus* that displays subtype selectivity for heteromeric voltage-gated $K^+$ (Kv1) channels [9,10]. The selectivity of this toxin has recently been employed to identify new subclasses of mechanosensory neurons — functionally distinct sensory neurons that detect mechanical stimuli and transmit the signal to the central nervous system (CNS) [11]. Somatosensory neurons comprise a heterogeneous population of neurons that can be divided into subclasses using constellation pharmacology [12]. Using this approach, individual neuronal cells in a population of mouse dorsal root ganglion (DRG) neurons are screened with a combination of calcium imaging and pharmacological compounds that each elicit a characteristic response used to differentiate neuronal cell types.

Conotoxins are produced and folded in the endoplasmic reticulum (ER) of cone snail venom glandular cells. Thus, conotoxin preproproteins typically comprise a signal sequence for entry into the ER, a propeptide region of largely unknown function, and the mature peptide that is proteolytically released from the propeptide [13]. Conotoxins are classified into gene superfamilies based on N-terminal signal sequence similarity [14], with more than 50 gene superfamilies identified to date [13]. Some superfamilies comprise several subfamilies (which we term "classes" in the current work). For instance, this is the case for the C-superfamily that comprises the consomatin and contulakin-G classes [15]. In contrast to the signal sequence, the mature peptide region exhibits remarkable sequence variability except for the presence of conserved cysteines that form disulfide bonds critical for stability. In addition to disulfide bonds, conotoxins can acquire a variety of other posttranslational modifications, such as C-terminal amidation, O-glycosylation, hydroxylation, and bromination that can, in some cases, influence target binding [16–19].

The advent of new sequencing technologies and bioinformatics tools for transcriptome analysis has revealed thousands of previously unknown animal venom peptide and protein sequences in recent years [20]. Although cone snails mostly express short peptide toxins (mean length of the mature peptide: 42 residues [13]), the many new available sequences reveal that they also produce larger toxins. These more complex molecules have not been comprehensively explored, mostly because of limitations in the production of large, cysteine-rich proteins. Specifically, unlike the short peptide toxins, the larger toxins are rarely amenable to

Protein Data Bank; PDI, protein-disulfide isomerase; RMSD, root mean square deviation; ROI, region of interest; RP-HPLC, reversed-phase high pressure liquid chromatography; SapA, saposin A; SAPLIP, saposin-like proteins; SAXS, small angle X-ray scattering; SVD, singular value decomposition; SDS-PAGE, sodium dodecyl sulfate-polyacrylamide gel electrophoresis; SLC, saposin-like conotoxin; SR, seal resistant; TEV, tobacco etch virus; Ub, ubiquitin; Ub-His$_{10}$, Ub containing 10 consecutive histidines; UTR, untranslated region.

chemical synthesis and subsequent in vitro folding. Here, we coin the term "macro-conotoxin" for this group of conotoxins generally longer than 50 amino acid residues.

In this study, we identify and investigate a previously uncharacterized conotoxin, Mu8.1, from the fish-hunting snail *Conus mucronatus*. We produce this unusually large conotoxin of 89 residues using a modified *Escherichia coli* expression system and uncover that it belongs to a distinct class of conotoxins evolutionarily related to the con-ikot-ikots as well as 2 hitherto undescribed conotoxin classes. We demonstrate that Mu8.1 inhibits depolarization-induced $Ca^{2+}$ influx in mouse peptidergic nociceptors, likely through targeting the voltage-gated R-type (Cav2.3) $Ca^{2+}$ channels. In addition to identifying a previously unrecognized conotoxin superfamily and providing structural insight at the atomic level, this study establishes the potential of Mu8.1 as a new scaffold for the investigation of the role of Cav2.3-mediated currents in sensory neurons. Our work also demonstrates that an understudied pool of macro-conotoxins, such as Mu8.1, is amenable to detailed structural and functional investigation.

## Results

### Mu8.1 defines a new class of conotoxins evolutionarily related to the con-ikot-ikots

Transcriptome analysis of the venom gland of *C. mucronatus*, a fish-hunting species from the *Phasmoconus* clade [20], revealed the presence of 2 previously unrecognized toxin transcripts. Their high degree of sequence similarity designated them to be allelic variants that we assigned the names Mu8.1 and Mu8.1ii [21]. The protein sequence of Mu8.1 displays the characteristic tripartite organization of conotoxins consisting of a predicted N-terminal signal sequence (21 residues), an intervening propeptide region (5 residues), and, lastly, the toxin-encoding region (89 residues) (Fig 1A). The Mu8.1 and Mu8.1ii sequences differed only in the substitution of 3 residues in the C-terminal region (Fig 1B).

We initially performed a BLASTp search [22] using the Mu8.1 precursor as the query sequence to find Mu8.1-related sequences from other *Conus* species. Three of the identified Mu8.1-related toxins, Vx65, Vx78, and Vx79 from *Conus vexillum*, have previously been identified at the protein level in venom extracts by MS/MS [23]. Several of the identified sequences were annotated as con-ikot-ikot toxins. However, sequence alignment with the original con-ikot-ikot toxin [24] suggested that Mu8.1 may belong to a different class of conotoxins. To identify a more complete set of conotoxins belonging to the same class as Mu8.1, we mined the transcriptome data available in the National Center for Biotechnology Information (NCBI), DNA Databank of Japan (DDBJ), and China National Genebank (CNGB) sequence repositories. Searching 602,377 assembled transcripts from these venom gland datasets (S1 Table), we identified a total of 104 sequences from 38 species. These sequences included examples from the snail-hunting species *Conus marmoreus*, *Conus textile*, and *Conus gloriamaris*, as well as several from various worm-hunting species. The identified signal sequences are highly conserved and harbor a MDMKMTFSGFVLVVLVTTVVG consensus sequence (S1 Fig). Several of the sequences contain 2 additional N-terminal residues, MT. All sequences contain 10 conserved cysteine residues in the region of the mature toxin with identical loop sizes between them, with a few sequences containing 1 or 2 additional cysteines. As is common in conotoxins, there is a high variability between the mature sequences. Still, 10 residues (in addition to the cysteines) are strictly conserved among species (S1 Fig). Based on the structural similarity with proteins carrying a saposin domain (see further below), we refer to this new class of toxins as saposin-like conotoxins (SLCs).

The database annotation of several SLCs as con-ikot-ikots led us to investigate a potential evolutionary relationship between these 2 classes. To do so, we identified all published *Conus*

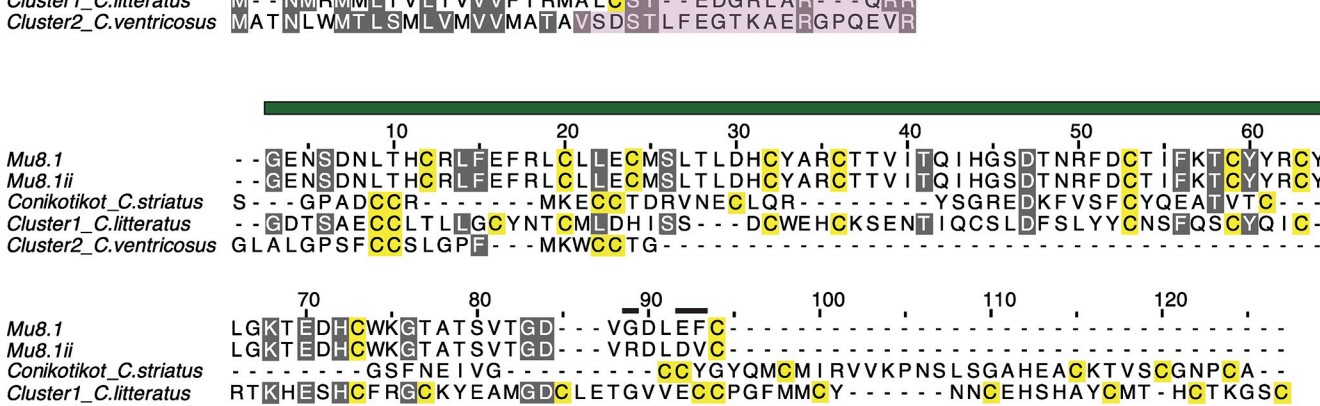

**Fig 1. Mu8.1 belongs to a new conotoxin superfamily. (A)** Mu8.1 prepro-sequence annotated with colored bars indicating the predicted tripartite organization. Mauve: signal sequence; pink: propeptide region; green: mature Mu8.1. Cysteine residues are colored red. **(B)** Alignment of Mu8.1/ii and select sequences from the con-ikot-ikot (from *C. striatus*), Cluster 1 (from *C. litteratus*), and Cluster 2 (from *C. ventricosus*) classes. Top: multiple sequence alignment of the predicted signal and propeptide sequences (mauve and pink bars denote signal and propeptide organization for Mu8.1, and the shaded pink boxes indicate the propeptide sequences of con-ikot-ikot, Cluster 1 and Cluster 2). Bottom: alignment of the mature toxin sequences (green bar). The 3 sequence differences between Mu8.1 and Mu8.1ii are marked with black bars above the residues in the alignment. Precursor sequences provided in S1 Data. Grey: columns with at least 60% sequence identity; yellow: cysteine residues.

sequences with sequence similarity to con-ikot-ikots (retrieved from GenBank) and grouped these using CLuster ANalysis of Sequences (CLANS) clustering analysis. This method uses all-against-all BLAST e-values to attract sequences with high similarity and repel sequences of little similarity, thereby forming clusters of highly similar sequences. The CLANS analysis (S2 Fig) showed 4 distinct but interconnected clusters corresponding to the identified toxin classes: the "classical" con-ikot-ikots, the SLCs, and 2 previously unrecognized conotoxin classes (Clusters 1 and 2). The numerous links interconnecting the 4 clusters suggest a common evolutionary origin. To further probe the putative evolutionary relationship between the sequences of the 4 classes, we compared the nucleotide sequences of their 5′ untranslated regions (UTRs) and the beginning of their open reading frames (ORFs) using randomly selected sequences from each class. This sequence alignment showed a high sequence identity of the 5′ UTR in the 4 toxin classes (S3 Fig), further suggesting a common evolutionary origin. The conclusion that the sequences of the 4 identified classes are related through an ancestral gene was corroborated by the finding that their gene structures are very similar, with all introns occurring in the same phase (phase 1) (S4 Fig). We concluded that the 4 identified classes—SLC, con-ikot-ikot, Cluster 1, and Cluster 2—together comprise a superfamily where the individual classes have related signal sequences, but distinct mature regions (Fig 1B).

## Recombinantly expressed Mu8.1 purifies as a single, fully oxidized species from *E. coli*

To gain insight into the structure and function of the new SLC family of conotoxins, we expressed Mu8.1 using the csCyDisCo system [25]. This system, based on the original CyDisCo system [26,27], allows disulfide-bond formation in the cytosol of *E. coli* as a result of expression of the Erv1p oxidase along with 2 protein disulfide isomerases (human protein-disulfide isomerase (PDI) and a conotoxin-specific PDI) from an auxiliary plasmid [25,28]. Mu8.1 was expressed as a fusion protein with an N-terminal ubiquitin (Ub) tag containing 10 consecutive histidine residues inserted into a loop region followed by a tobacco etch virus (TEV)-protease recognition site (S5A Fig). The beneficial effect of the csCyDisCo system was confirmed by SDS-PAGE gel and western blot analysis, where the Ub-His$_{10}$-Mu8.1 fusion protein was found in the soluble fraction (to approximately 50%) only when coexpressed with Erv1p and the 2 PDIs (Fig 2A), as we have previously observed for other conotoxins [25,28]. TEV protease cleavage of Ub-His$_{10}$-Mu8.1 resulted in the liberation of the 89-residue mature conotoxin, which was purified (>95% purity; Fig 2B) in a final yield of approximately 1 mg per liter of culture.

Disulfide-bond formation in Mu8.1 was verified by SDS-PAGE analysis and full oxidation by MALDI-TOF mass spectrometry. Upon reduction, a clear mobility shift was observed by SDS-PAGE (Fig 2B), and MALDI-TOF mass spectrometry of reversed-phase high pressure liquid chromatography (RP-HPLC)-purified Mu8.1 verified the presence of a single species with a mass of 10,181.7 Da (S5B Fig), a value that fits the theoretical average mass of fully oxidized Mu8.1 of 10,181.5 Da.

## Mu8.1 is an α-helical, dimeric protein in solution

More detailed insight into the structural properties of Mu8.1 in solution was obtained using circular dichroism (CD) spectroscopy, analytical gel filtration, and small-angle X-ray scattering (SAXS) measurements.

The CD spectrum of Mu8.1 showed characteristic features of an α-helical structure with minima at 208 nm and 218 nm, and a global maximum at 195 nm (S5C Fig). To probe the quaternary structure of Mu8.1 in solution, we first performed analytical size exclusion chromatography on Mu8.1. Regardless of the concentration (1 μM or 100 μM), Mu8.1 eluted as a single, symmetric peak with an apparent molecular weight of 22 to 24 kDa, suggesting a dimer under these conditions (Fig 2C).

To corroborate this result, SAXS measurements were carried out at 5 protein concentrations ranging from 1 mg/mL to 13.2 mg/mL (100 μM to 1.3 mM). Here, at low q-values, we observed an increase in the intensity with increasing protein concentration, indicating an increase in size (S6 Fig and Table A in S2 Table). A dimensionless Kratky plot of the data revealed a shift in peak position with increasing concentration, suggesting a change from a spherical conformation concomitant with increasing concentration (Fig 2D). Singular value decomposition was employed to assess the number of species present, followed by calculation of their relative volume fractions (Table B in S2 Table). The results revealed the presence of 3 oligomeric species, with the dominant being a dimer (85% at 1 mg/mL) and increasing occurrence of tetrameric and hexameric species that correlated with increasing concentration. No monomer was observed.

Based on evidence from the analytic gel filtration and SAXS measurements, we concluded that Mu8.1 exists primarily as a dimer in solution.

## The crystal structure of Mu8.1

To elucidate key structural features of the SLC superfamily, we determined crystal structures of Mu8.1 from 2 different crystal conditions referred to as Mu8.1_38 and 59 (see

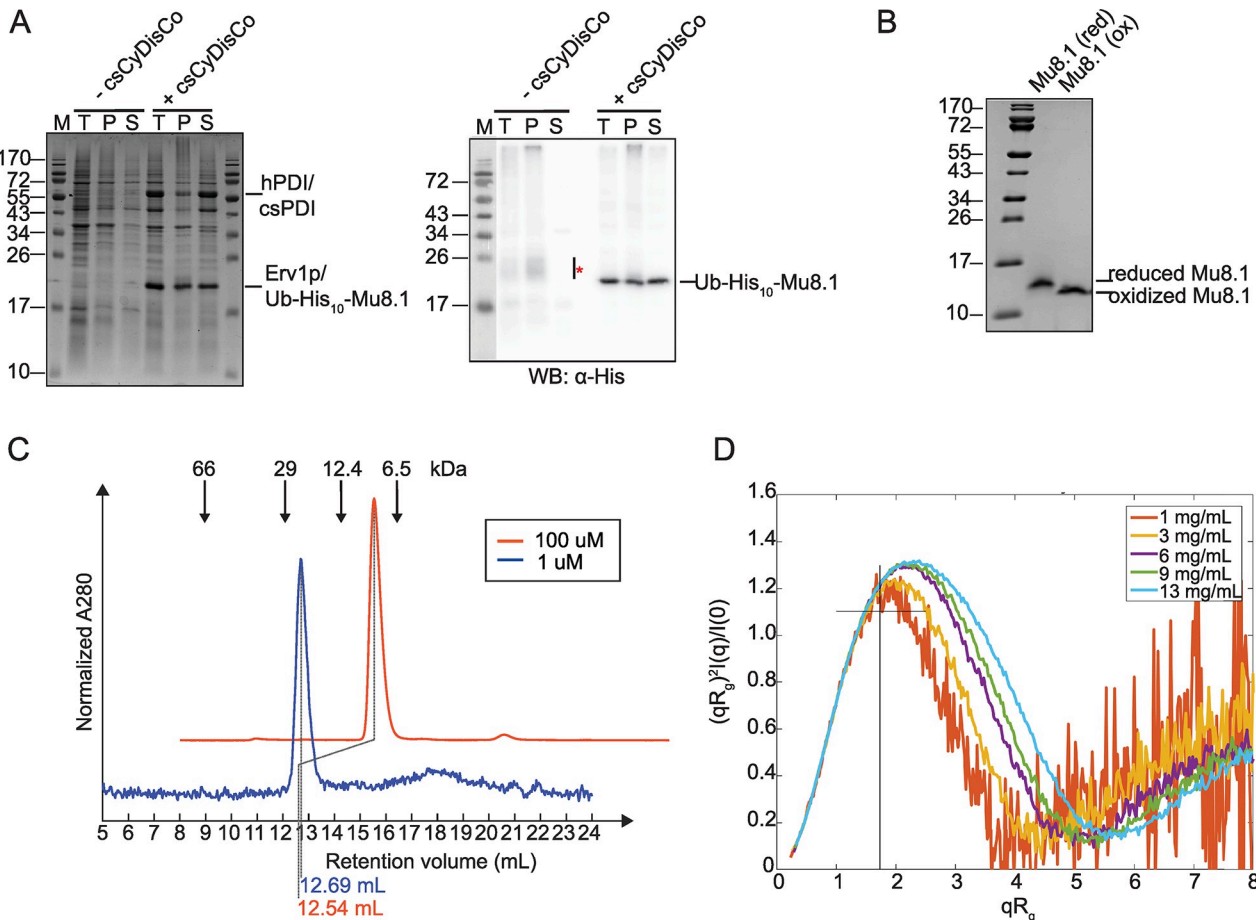

**Fig 2. Recombinantly expressed Mu8.1 exists primarily as a noncovalent dimer in solution.** (**A**) Comparison of Ub-His$_{10}$-Mu8.1 expressed in the absence (−csCyDisCo) or presence (+csCyDisCo) of the csCyDisCo expression system in *E. coli* BL21(DE3) cells. Left panel: 15% SDS-PAGE gel stained with Coomassie Brilliant Blue showing the molecular weight marker (M), the total cell extract (T), resuspended pellet after lysis and centrifugation (P), and the soluble fraction (S) from cells expressing Ub–His$_{10}$–Mu8.1. Note that Ub-His$_{10}$-Mu8.1 and Erv1p migrate similarly. Protein levels are directly comparable between lanes. Right panel: western blot probed with an anti-His antibody (α-His) for detection of Ub-His$_{10}$-Mu8.1 in the same samples loaded on the SDS-PAGE gel. The line labeled with a red asterisk indicates the migration of what is likely reduced and partially oxidized species of Ub-His$_{10}$-Mu8.1. Original SDS-PAGE gel and western blot images provided in S1 Raw Images. (**B**) 15% Tris-Tricine SDS-PAGE gel analysis of purified Mu8.1 in the reduced (40 mM DTT; red) and oxidized (ox) state. Original SDS-PAGE gel and western blot images provided in S1 Raw Images. (**C**) Analytic gel filtration of Mu8.1 at 100 μM (orange) and 1 μM (blue) concentrations. Arrows denote the elution volumes of standard proteins relating to the blue trace. The dotted lines show the retention volumes of Mu8.1 in each sample. The peak intensity of the 2 samples was min-max normalized to allow comparison, and the orange plot was offset for clarity. (**D**) Normalized Kratky plot from small angle X-ray scattering analysis of 5 concentrations of purified Mu8.1 (1 mg/ml, orange; 3 mg/ml, yellow; 6 mg/mL, purple; 8 mg/mL, green; 13 mg/mL, light blue). The cross denotes the peak position of a globular protein. Source data for quantifications provided in S2 Data.

Materials and methods) at resolutions of 2.33 Å and 2.1 Å, respectively (Fig 3). The asymmetric unit of Mu8.1_59 accommodates 6 molecules that form 3 equivalent dimers (S7 Fig). The asymmetric unit of Mu8.1_38 includes 2 molecules that form a single dimer with an interface equivalent to the one present in the 3 dimers in Mu8.1_59. Each protomer of the Mu8.1 structure comprises 2 regions (Fig 3A and 3B). The first region comprises an N-terminal 3$_{10}$-helix (3$_{10}$N), followed by 2 α-helices (α1 and α2), and a 3$_{10}$-linker helix (3$_{10}$L). The second region comprises 2 α-helices (α3 and α4), and the C-terminal 3$_{10}$-helix (3$_{10}$C). The disulfide bonds are found primarily within each of these 2 regions. Within the first region, the Cys18-Cys34 and Cys22-Cys30 disulfide bonds connect α1 and α2. Further, the 3$_{10}$N-helix connects to α3 through the Cys10-Cys51 disulfide bond. In the second region, Cys61-Cys71 links α3 and α4, and Cys57-Cys89 tethers the C-terminus to α3 (Fig 3B).

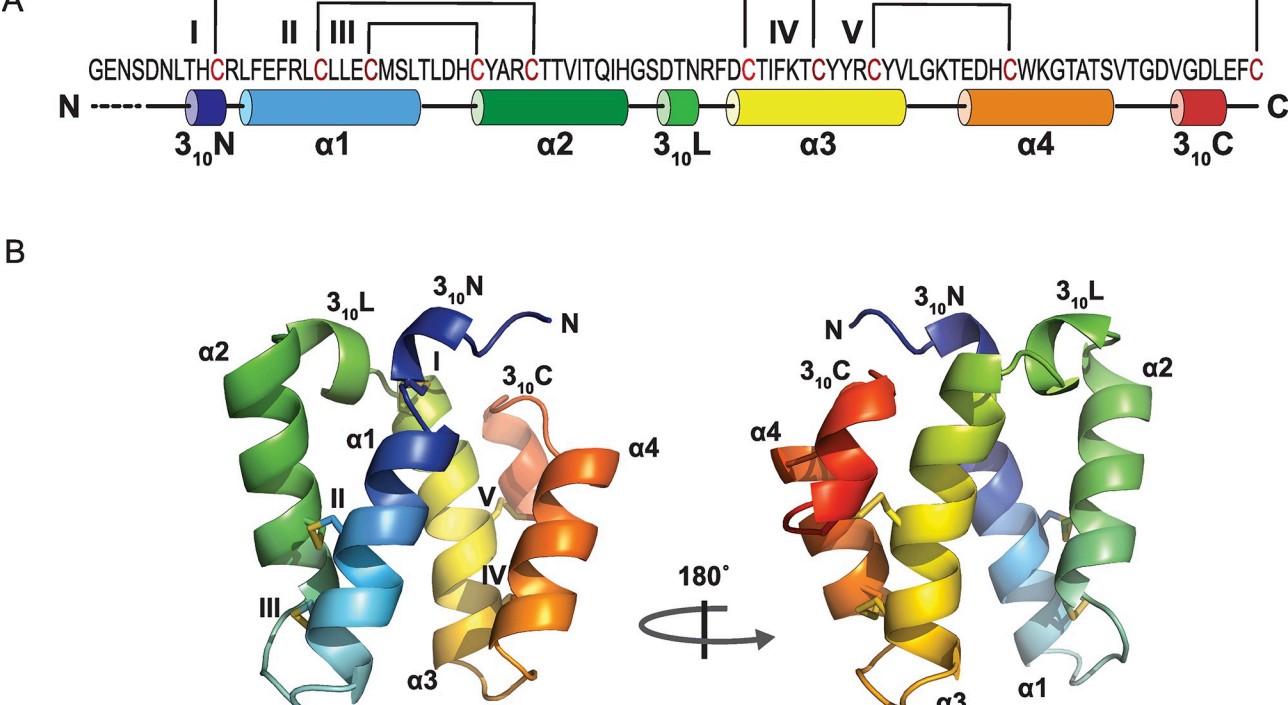

**Fig 3. Crystal structure of Mu8.1. (A)** Graphic representation of the Mu8.1 structure with disulfide bridges represented by brackets (numbered by Roman numerals) and α- and $3_{10}$-helices represented by cylinders (numbered by Arabic numerals). Four amino acid residues at the N-terminus not resolved in the crystal structure are shown as dotted lines. **(B)** Cartoon representation of the Mu8.1 protomer containing three $3_{10}$ helices interspersed with 4 α-helices labelled as α1-α4. The disulfide bridges are shown as yellow stick models and labelled with Roman numerals as in Panel A.

The 2 helical regions surround a hydrophobic core predominantly formed by aromatic residues (Phe15, Tyr31, Tyr58, Tyr59, Trp72; Fig 4A). Moreover, α1, α3, and α4 are held together by a network of contacts between Arg16, Tyr58, Glu68, and Trp72, where ionic interactions are present between the side chains of Arg16 and Glu68, whereas Tyr58 and Trp72 partake in T-shaped pi–pi interactions (Fig 4A). This network of residues is completely or highly conserved among the SLC superfamily toxins (S1 Fig). The only variations are seen in 4 sequences where Arg16 is substituted with a Lys and Tyr58 is substituted by a Phe.

The dimer interface is characterized by both hydrophobicity as well as the presence of a water-filled cavity surrounded by the polar residues Thr36, Asn47, Thr52, and Thr56 (Fig 4B). The residues forming the largely dry (water-excluded) hydrophobic interface include Tyr31, Ala32, Phe49, Ile53, Tyr59, and Val83. While the residues involved in structural stabilization of the Mu8.1 monomer are highly conserved among SLC superfamily proteins, the amino acid residues comprising the dimer interface exhibit a higher degree of variability (S1 Fig). A potential functional role of 4 additional conserved residues without any apparent structural purpose —Lys55, Arg60, Lys66, and His70—remains speculative at present but could represent residues that interact with the molecular target of this toxin family.

## Mu8.1 has structural similarity with con-ikot-ikot but does not modify GluA2 desensitization

Structural similarity often correlates with similar functional properties and, consequently, may provide important insight into protein function. We, therefore, searched the entire Protein Data Bank (PDB) for structures exhibiting similarity to Mu8.1. The search identified 48 hits

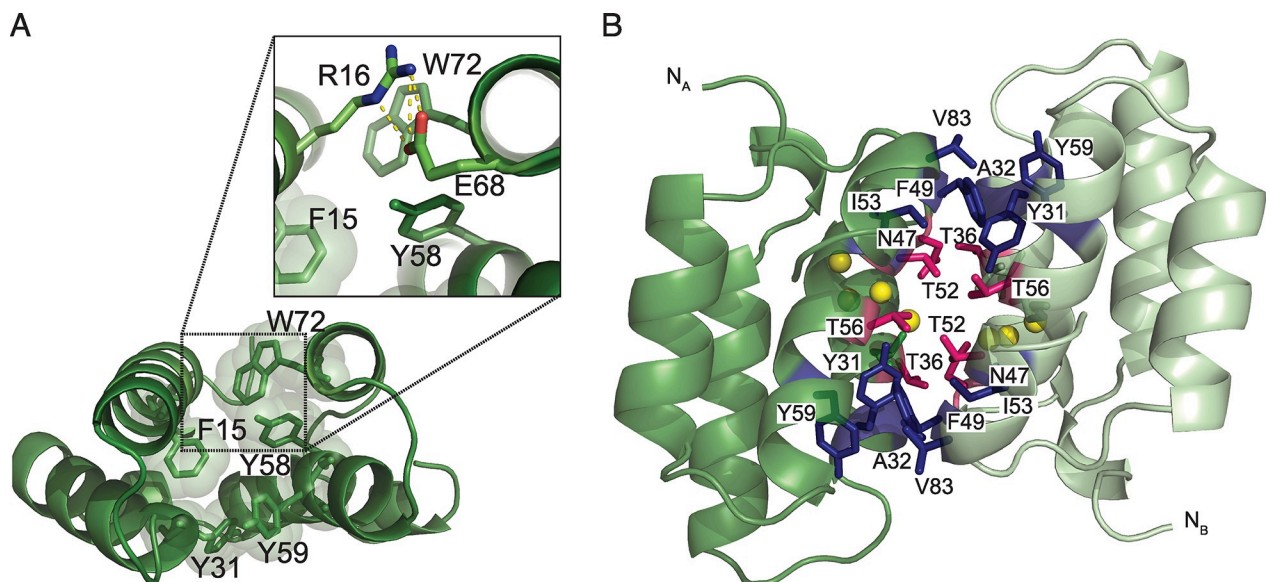

**Fig 4. Crystal structure of dimeric Mu8.1. (A)** Cartoon representation of the Mu8.1 protomer, with the aromatic residues (F15, Y31, Y58, Y59, and W72) contributing to the hydrophobic core depicted as stick models and the corresponding van der Waals radii depicted as translucent spheres. Zoom: Select conserved amino acid residues—R16, Y58, E68, and W72 (shown as stick models)—whose interactions play a structural role (see main text for details). Ionic interactions are shown as yellow dotted lines. **(B)** "Side view" of the Mu8.1 dimer highlighting the amino acid residues (depicted as sticks) composing the dimer interface. Protomer A is shown in forest green, and protomer B in pale green. Hydrophobic residues are in dark blue, and polar residues in pink. Water molecules are represented by yellow spheres.

showing a high degree of similarity to the Mu8.1 protomer structure, with most structures displaying a saposin-like fold (see below). Notably, con-ikot-ikot from *Conus striatus* also showed structural similarity with Mu8.1. The schematic overview of Mu8.1, con-ikot-ikot, and human saposin A (SapA) in Fig 5A reveals the similar architecture of these proteins.

Con-ikot-ikot targets the AMPA-type of ionotropic glutamate receptors and has been shown to inhibit desensitization of the receptor producing a sustained agonist receptor-mediated current [24,30]. The crystal structure of con-ikot-ikot, determined in complex with the GluA2-type AMPA receptor, has shown a homo-dimeric protein covalently linked by disulfide bonds formed between 3 of the 13 cysteine residues present in each monomer [29] (Fig 5A). Despite low sequence similarity between con-ikot-ikot and Mu8.1 (Fig 1B), the structures of the Mu8.1 protomer and con-ikot-ikot superimpose well with a root mean square deviation (RMSD) of 2.14 Å for all Cα atoms (Fig 5B). Nevertheless, despite the high degree of structural similarity between the 2 proteins, Mu8.1 failed to modulate the desensitization of the GluA2 AMPA receptor transiently transfected into HEK293 cells (S8 Fig).

## Mu8.1 and con-ikot-ikot display a saposin-like fold

Saposin-like proteins (SAPLIPs) comprise a protein family with over 200 members that perform a variety of biological functions and are found in a phylogenetically diverse group of eukaryotes. Most of these involve lipid interaction and lead to local disordering of the lipid structure, membrane perturbation, or membrane permeabilization [31], although the fold is also associated with other functions [32]. SAPLIPs display a characteristic 4-helix bundle structure stabilized by a conserved pattern of 3 disulfide bonds (Fig 5A, 5C and 5D).

SapA was one of the top hits in our search, although the SapA and Mu8.1 sequences only share 21.6% sequence identity. The SapA and Mu8.1 have similar structural topologies and superimpose with an RMSD of 1.81 Å for all Cα atoms, with the 2 first α-helices

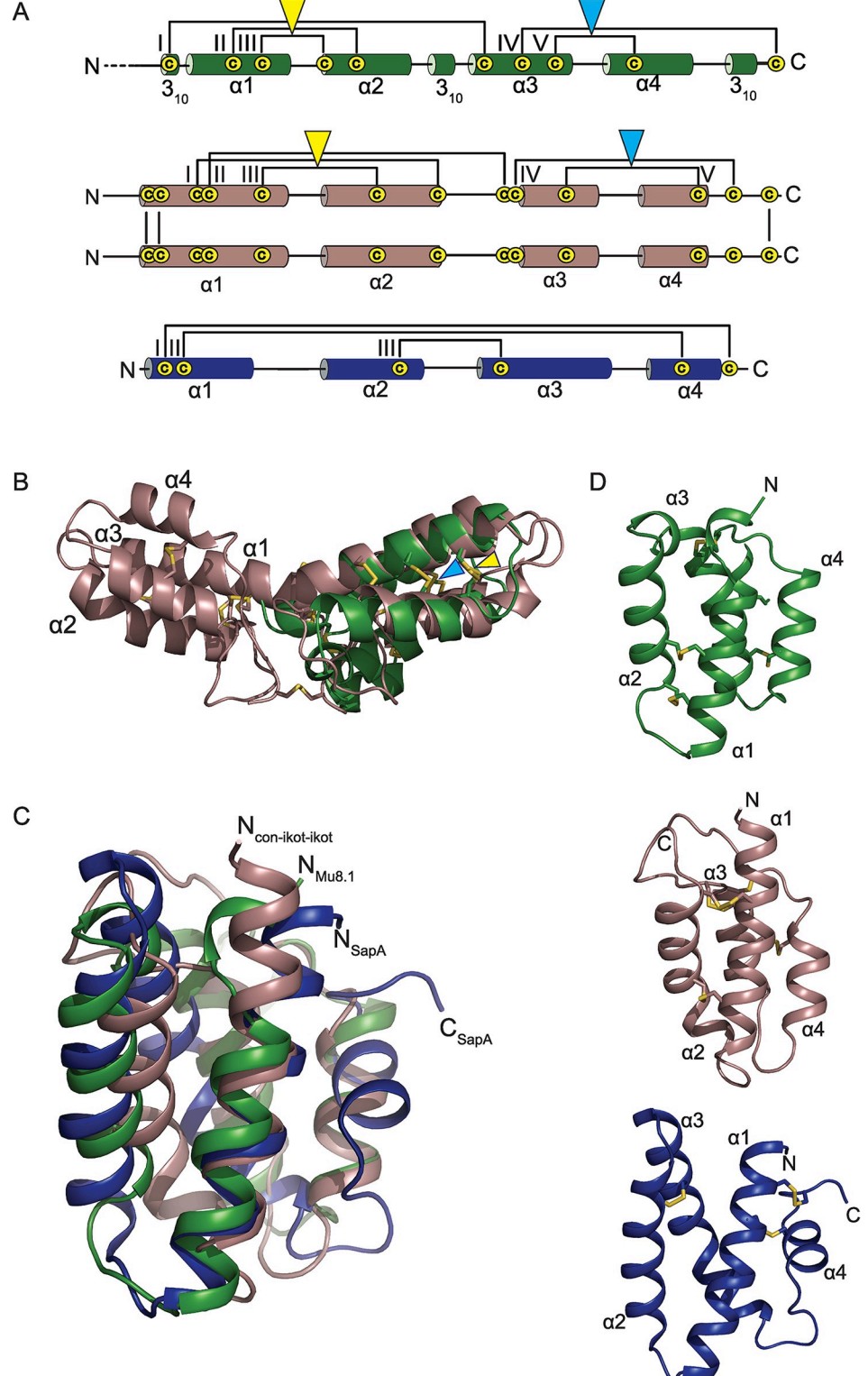

**Fig 5. Monomeric Mu8.1 exhibits a saposin-like fold. (A)** Graphic representations of Mu8.1 (green), con-ikot-ikot (dirty violet), and SapA (blue). Con-ikot-ikot is a homodimer held together by 3 intermolecular disulfide bonds, with each subunit comprising 4 α-helices and 5 disulfide bonds [29]. SapA comprises 4 α-helices and 3 disulfide bonds. Disulfide bonds are depicted as brackets and labelled with Roman numerals. Amino acid residues not resolved in the crystal structure of Mu8.1 are shown as dotted lines. The yellow and light blue triangles denote structurally conserved

disulfide bonds between Mu8.1 and con-ikot-ikot. (**B**) Superimposition of the structures of monomeric Mu8.1 (forest green) and con-ikot-ikot dimer (PDB:4U5H) (dirty violet) are shown in cartoon representation. Disulfide bonds are represented as yellow sticks. The superimposition was performed using the CLICK server as described in the Materials and methods section and visualized in Pymol. (**C**) Overlay of cartoon representations of monomeric Mu8.1 (forest green), monomeric con-ikot-ikot (dirty violet), and SapA (PDB: 2DOB) (blue). Con-ikot-ikot and Mu8.1 superposition was performed with the CLICK server as described in Materials and methods. SapA was manually superimposed with the 2 other molecules in Pymol using the sequence-independent program "super". (**D**) Mu8.1 (green), con-ikot-ikot (dirty violet), and SapA (blue) shown separately in the same orientation as in Panel C.

superimposing especially well (Fig 5C). A significant difference is constituted by the disulfide pattern in the 2 proteins (Fig 5A and 5D). The functional implications of this observation suggest that Mu8.1 is unlikely to be involved in lipid binding (see Discussion). Overall, this analysis revealed a previously undiscovered structural resemblance between con-ikot-ikot and SAPLIPs that extends to Mu8.1.

## Mu8.1 modulates depolarization-induced $Ca^{2+}$ influx in sensory neurons

To assess the bioactivity of Mu8.1, we next tested the effects of Mu8.1 on $Ca^{2+}$ influx in somatosensory DRG neurons that relay sensory input to the CNS. These cells were isolated from transgenic reporter mice in which the regulatory elements of the calcitonin gene-related peptide (CGRP) drive the expression of green fluorescent protein (GFP), whereby GFP labeling identifies peptidergic nociceptive neurons. Overall, 7 major classes of sensory neurons responsible for processing the sensations of cold, heat, mechanical cues, and pain were assigned based on cell size, GFP labeling, and isolectin B4 staining [33], as well as their responses to mustard oil (AITC), menthol, capsaicin, and conotoxin RIIIJ, according to the functional classification of somatosensory neurons proposed by Giacobassi and colleagues [11].

Representative examples from neurons belonging to all somatosensory subclasses assayed are shown in Fig 6A. DRG neurons incubated in observation solution (4 mM KCl) were sequentially depolarized by pulses (15 seconds) of high potassium (25 mM KCl) extracellular solution, which caused stereotypical rises in intracellular $Ca^{2+}$ concentration evidenced by the increase in Fura-2 signal. Incubation with 10 μM Mu8.1 produced a decrease in subsequent $Ca^{2+}$ peaks elicited by the high $K^+$ stimulus in 20.3 ± 3.0% of the neurons analyzed ($n = 2,365$; Fig 6B pie chart), predominantly in peptidergic nociceptors (68% of all affected cells, green bar Fig 6B). In addition to the peptidergic nociceptors, 10 μM Mu8.1 reduced the calcium signals of DRG neurons identified as large-diameter mechanosensors (12.6% of all affected cells) and C-low threshold mechanoreceptors (C-LTMRs; 7% of all affected cells) as shown in Fig 6B (magenta and lilac bars, respectively).

Mu8.1 inhibitory effects on cytosolic $Ca^{2+}$ concentration was estimated after min-max normalization (see Materials and methods) and evaluated by two-tailed $t$ tests (Fig 6C). Mu8.1 (10 μM) significantly curtailed the total influx of $Ca^{2+}$ elicited by KCl application in peptidergic nociceptors (f(x) = −0.14 ± 0.01, $n = 596$; $p < 0.001$), C-LTMRs (f(x) = −0.06 ± 0.01, $n = 190$; $p < 0.001$), and large-diameter mechanosensors (f(x) = −0.03 ± 0.01, $n = 48$; $p = 0.0279$).

## Mu8.1 inhibits recombinant and native Cav2.3-mediated currents

Given the ability of Mu8.1 to inhibit cellular $Ca^{2+}$ influxes in DRG neurons, we assessed the modulatory effects of Mu8.1 over a comprehensive panel of recombinant voltage-gated calcium, sodium, and potassium channels (Cav, Nav, and Kv, respectively) by automated patch clamp (APC) electrophysiology as well as a large array of GPCRs by high-throughput

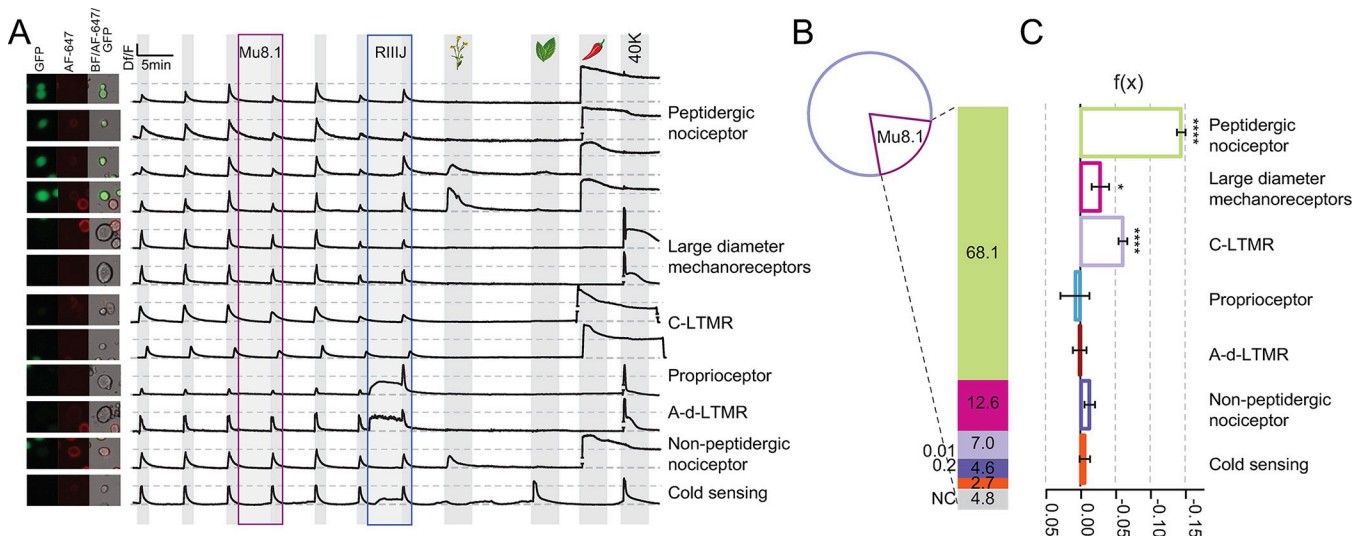

**Fig 6. Mu8.1 inhibits calcium entry into mouse sensory neurons. (A)** Intracellular calcium changes in response to sequential pharmacological treatments from 7 classes of DRG neurons (labelled on the right). Each trace represents the calcium signal (ΔF/F) of the neuron pictured on the left (GFP-CGRP+: peptidergic nociceptors; Alexa Fluor 647-Isolectin B4+: nonpeptidergic nociceptors, and bright-field). KCl depolarization pulses (25 mM) are indicated by light grey shading. In the presence of Mu8.1 (10 μM, mauve box) KCl-induced calcium peaks are reduced in peptidergic nociceptors (traces 1–4), large-diameter mechanoreceptors (traces 5–6), and C-LTMRs (traces 7–8). Class-defining pharmacology: RIIIJ (1 μM, blue box); AITC (100 μM; mustard flower), menthol (400 μM; peppermint leaf), and capsaicin (300 nM; chili pepper). A pulse of KCl (40 mM) was used to elicit maximum calcium signal at the end of the experiment. Horizontal lines flanking breaks within a trace signify graphical adjustment of trace amplitude to avoid overlap of neighboring traces. **(B)** Pie chart representing the population of neurons analyzed ($n = 2,365$), highlighting the 20.3 ± 3% of Mu8.1 (10 μM)-sensitive sensory neurons. The bar graph provides the percentage of each class of Mu8.1-sensitive DRG neurons. **(C)** Quantification of the relative change f(x) in KCl-induced intracellular calcium signal observed in the presence of Mu8.1 (10 μM) (see Materials and methods for normalization details). Two-tailed $t$ test * $p \leq 0.05$; **** $p < 0.0005$. Source data for individual traces and quantifications found in S3 Data. AITC, allyl isothiocyanate; CGRP, calcitonin gene-related peptide; C-LTMR, C-low threshold mechanoreceptor; DRG, dorsal root ganglia; GFP, green fluorescent protein.

screening assays. Among all the recombinant channels tested (S3 Table), Mu8.1 exhibited the highest potency against Cav2.3 channels. Fig 7 shows representative current traces of human Cav2.3-mediated whole-cell currents exposed to increasing concentrations of Mu8.1 recorded from HEK293 cells (Fig 7A) and the resulting concentration–response curve rendering an IC$_{50}$ of 5.8 μM and a Hill coefficient close to unity (0.97) (Fig 7B). Mu8.1 reversibly inhibited Cav2.3 peak currents (S9 Fig) without significantly affecting the voltage dependence of activation for the channel (S10 Fig).

At concentrations above 30 μM, Mu8.1 displayed weak inhibitory effects over Cav2.1, Cav2.2, Cav3.1–3, and Kv1.1 channels with near equipotency (IC$_{50}$s calculated from inhibition at a single concentration of Mu8.1 are summarized in S3 Table), whereas no obvious modulatory effects over heterologously expressed Cav1.2, Nav1.4, Nav1.7, hERG, Kv1.2, Kv1.3, or Kv4.3 channels were noted (S3 Table). The same was observed for a large panel of recombinant GPCRs and other receptors assessed through β-arrestin recruitment and radioligand binding assays (S11 Fig). Overall, our comprehensive functional screening identified the R-type Cav2.3 channel as the highest affinity mammalian target of Mu8.1 (although inhibition occurs with modest potency). This finding aligns with the observed effects of Mu8.1 on DRG subpopulations, including peptidergic nociceptors and C-LTMRs, which are known to express abundant levels of Cav2.3 channels [34–36].

Using constellation pharmacology, we demonstrated that Mu8.1 inhibition of DRG neuron calcium signals were largely reversible shortly after removal of the peptide as well as concentration dependent. Reversibility of Mu8.1 actions can be surmised from the grey shading after Mu8.1 application in Figs 6A and 7C, while 3 μM and 10 μM Mu8.1 inhibited 32 ± 10% and

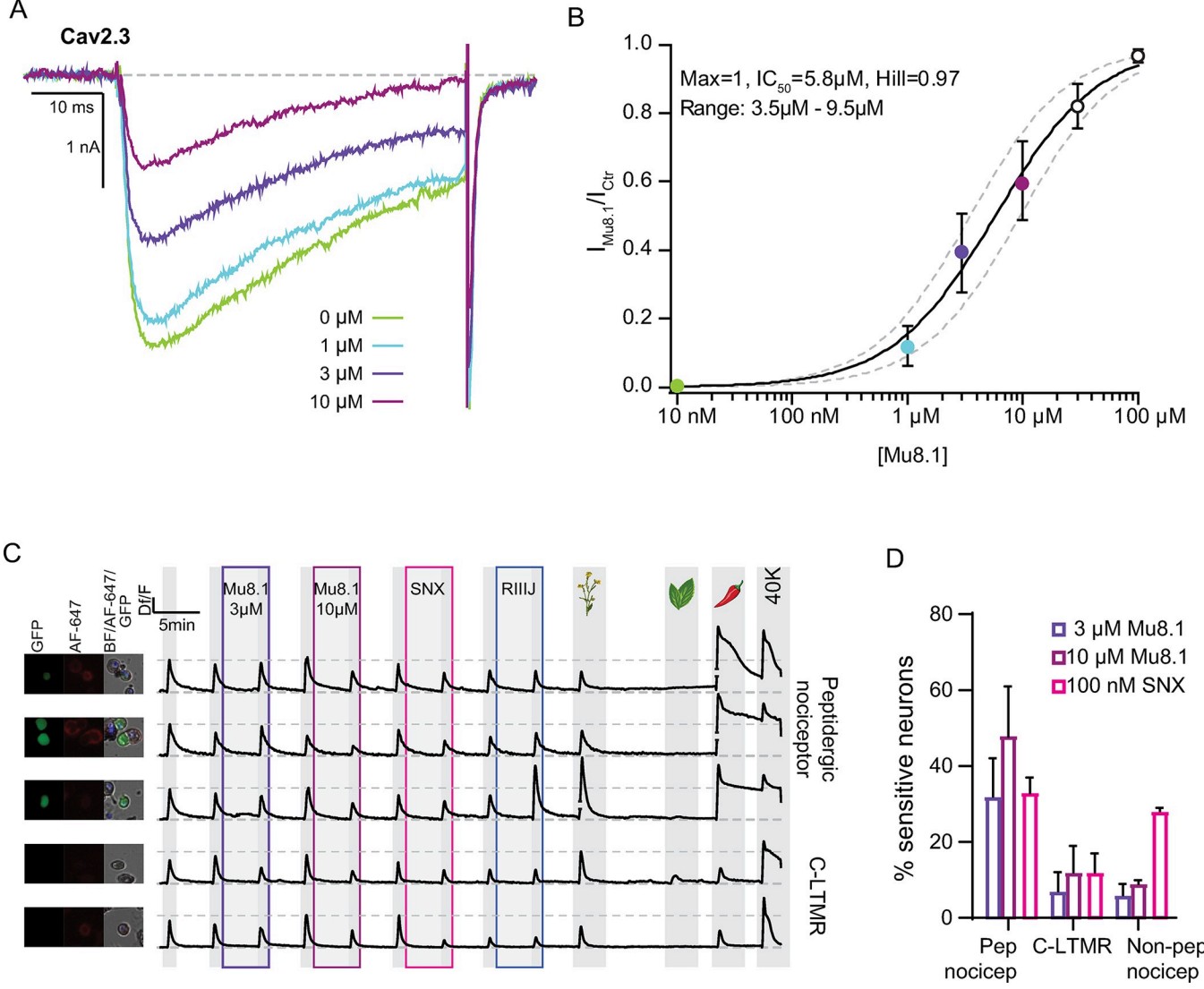

**Fig 7. Mu8.1 inhibits recombinant human Cav2.3-mediated currents in a concentration-dependent manner. (A)** Representative whole-cell Cav2.3-mediated currents in control (green) and in the presence of 1 μM (blue), 3 μM (purple), and 10 μM (mauve) Mu8.1. Depolarization-activated currents were elicited by a 50-ms test pulse to −10 mV (Vh −80 mV; 0.1 Hz). **(B)** Concentration–response relationship for the inhibition of Cav2.3 peak currents ($IC_{50}$ = 5.8 μM; nH: 0.97; $n$ = 5). Source data showing the recorded currents and quantifications provided in S4 Data. **(C)** Mu8.1 and SNX-482 target overlapping populations of peptidergic nociceptors and C-LTMRs. Each trace represents the calcium signal (ΔF/F) of the neuron pictured on the left (GFP-CGRP+: peptidergic nociceptors; Alexa Fluor 647-Isolectin B4+: nonpeptidergic nociceptors, and bright-field). Cells were treated with 3 μM Mu8.1 (purple box), 10 μM Mu8.1 (mauve box), and 100 nM SNX-482 (pink box). All other labels as in Fig 6A. Horizontal lines flanking breaks within a trace signify graphical adjustment of trace amplitude to avoid overlap of neighboring traces. **(D)** Relative quantification of peptidergic nociceptors (pep nocicep), C-LTMRs, and nonpeptidergic nociceptors (non-pep nocicep) sensitive to 3 μM and 10 μM Mu8.1 (purple and mauve, respectively) and 100 nM SNX-482 (pink). Percentages were calculated from the total number of cells recorded per class. Bars represent mean ± SD from 2 independent DRG isolations (190 and 810 neurons per group). Source data of individual traces and quantifications found in S4 Data. CGRP, calcitonin gene-related peptide; C-LTMR, C-low threshold mechanoreceptor; DRG, dorsal root ganglia; GFP, green fluorescent protein.

48 ± 13% of subsequent calcium peaks in peptidergic nociceptors, respectively (Fig 7C and 7D). Although overall reversibility was observed, 7.8% of peptidergic nociceptors did not recover after several full-bath media exchanges (S12 Fig).

SNX-482 is a spider venom peptide commonly used to study Cav2.3-mediated currents in native tissues [37], and, in contrast to Mu8.1, it is also a potent inhibitor of the Kv4 family of voltage-gated potassium channels [38]. We applied SNX-482 (100 nM) to DRG neurons after

Mu8.1 inhibition was reversed, verifying that these 2 venom-derived peptides target overlapping populations of somatosensory neurons, predominantly peptidergic nociceptors, and C-LTMRs (Fig 7C and 7D). Interestingly, SNX-482 also decreased $Ca^{2+}$ signals from approximately 28% of nonpeptidergic nociceptors. These results strongly indicate that the decrease in $Ca^{2+}$ influx observed upon exposure to Mu8.1 is mechanistically supported by the inhibition of Cav2.3 channels. In DRG neurons, SNX-482 not only reduced depolarization-induced $Ca^{2+}$ signals via inhibition of Cav2.3 channels but also displayed direct and indirect $Ca^{2+}$ signal amplification effects (S13 Fig) akin to the inhibition of voltage-gated $K^+$ conductances likely mediated by Kv4 potassium channel isoforms that are abundantly expressed in these neurons [39].

## Discussion

The vastly increased number of conotoxin sequences obtained in recent years from transcriptome sequencing constitutes a rich source of peptides for biomedical exploration. However, the production of new peptides often represents a bottleneck for their exploration—in particular, the characterization of large, disulfide-rich venom components is lagging. In this study, we identify the macro-conotoxin Mu8.1 from *C. mucronatus* as the founding member of the new SLC class and use a wide range of biochemical, biophysical, structural, and electrophysiological methods to provide a comprehensive characterization of the protein. Moreover, we uncover an unexpected evolutionary relationship between the SLCs and con-ikot-ikots that extends to 2 previously unrecognized toxin classes with all 4 thus defining a single conotoxin superfamily.

The relatively straightforward chemical synthesis of small peptides together with difficulties associated with the production of large venom components, in particular without a priori knowledge about the disulfide pattern, have biased functional studies towards small, disulfide-bridged conotoxin peptides shorter than 30 amino acid residues in length. The successful production of fully oxidized and correctly folded Mu8.1 in the csCyDisCo *E. coli* system highlights the feasibility of systematically exploring much larger multidisulfide conotoxins than was previously possible. Moreover, we recently developed the DisCoTune system based on CyDisCo to allow titration of T7 RNA polymerase repression [28]. This feature permits optimization of expression conditions to potentially further increase yields by fine-tuning the expression level of the (disulfide-rich) target protein to better match the level of the helper proteins (Erv1p and PDI).

With a few notable exceptions, such as con-ikot-ikot and proteins belonging to common toxin families like conkunitzins, metalloproteases, hyaluronidases, and Phospholipase $A_2$s [40–43], macro-conotoxins are generally unexplored. The large size of Mu8.1 and its modest potency against mammalian Cav2.3 channels prompts questions regarding the evolutionary advantage that producing and deploying this peptide may convey to *C. mucronatus*. In general, the size of macro-conotoxins likely confers specific properties not available to small toxins. For instance, larger toxins could participate in multivalent interactions with their targets, as noted for con-ikot-ikot and recently pointed out for bivalent venom peptides containing 2 homologous domains connected by an interdomain linker [44]. It is conceivable that noncovalent dimers, as seen in Mu8.1, could also allow interaction with, for instance, 2 identical subunits of a molecular target. Even in a monomeric state, large toxins may interact with different target subunits, whereas their larger binding interface may well provide higher target specificity.

We found that Mu8.1 is structurally similar to con-ikot-ikot and that both display a saposin-like fold. This result raised the possibility that Mu8.1 may perform a function involving lipid interactions. Human SapA and its homologs SapB, SapC, and SapD are small, nonenzymatic proteins required to break down glycosphingolipids within the lysosome [45]. In the

absence of lipid, SapA adopts a characteristic monomeric, closed conformation where α1 and α4 (held together by 2 disulfides) form the stem, and α2 and α3 (connected by 1 disulfide bond) form a hairpin region (Fig 5A and 5D). In the presence of lipids, SapA opens to expose a concave, hydrophobic surface for lipid binding. The primary areas of rearrangement are the loops between α1/α2 and α3/α4 that together operate as a hinge [46,47]. However, in contrast to SapA, the Mu8.1 structure is highly constricted by the network of disulfide bonds that cross-links the molecule (Fig 5A). Consequently, Mu8.1 is not likely to undergo a large conformational change to adopt an open conformation and, therefore, probably does not function in lipid binding in the same manner as the SAPLIPs.

Next, we sought to investigate if Mu8.1 and related sequences may have evolved from an endogenous saposin domain-containing protein. Despite using a combination of multiple BLAST algorithms and hidden Markov models of whole toxin sequences, individual exons, introns, and 5′ and 3′ UTRs against published genomes and transcriptomes from several non-venom-producing tissues, we did not identify a good candidate for an ancestral, endogenous gene. Likewise, an analysis of known *Conus* proteins harboring a saposin-like domain (S14 Fig) did not provide an obvious candidate for an ancestral endogenous gene. None of the endogenous proteins shared significant sequence similarities or gene structures with Mu8.1 or its related sequences. While we were unable to elucidate the evolutionary origin of Mu8.1, we find it probable that the SLCs (and the other 3 toxin classes) evolved from a common ancestral gene and then diverged to a point where their similarities are only apparent in their signal sequences, gene structures, and possibly also in their overall fold. As detailed above, Mu8.1 and con-ikot-ikot both display a saposin-like fold. Notably, AlphaFold structure predictions of Cluster 1 sequences show that these proteins are likely to also contain a saposin-like domain (S15 Fig). The Cluster 2 sequences are too short to encode a saposin-like domain. Therefore, we speculate that the ancestral gene from which the 4 classes evolved harbored a saposin-like domain, which has been retained in at least 3 of these classes throughout evolution.

Alternatively, although we find this less likely, the Mu8.1 and con-ikot-ikot sequences may have emerged through convergent evolution. If so, the saposin fold could constitute a "privileged" scaffold that has been selected during evolution due to favorable properties, such as structural stability and the ability to accommodate sequence variation. Similar privileged scaffolds are found in diverse toxin peptides and proteins that are functionally unrelated, including the inhibitor cystine knot, granulin, defensin, and Kunitz folds [25,48,49]. These examples demonstrate that irrespective of the evolutionary mechanism involved, the same fold can be repurposed to interact with diverse targets.

Functionally, neither the monomeric nor the dimeric states of Mu8.1 seem compatible with AMPA-receptor binding, thus rationalizing the observed lack of Mu8.1 effects on GluA2 desensitization (S8 Fig). First, the GluA2-binding surface of con-ikot-ikot corresponds to the dimer interface of Mu8.1. Second, 3 of the con-ikot-ikot residues shown to be important for GluA2 binding (Gln37, Glu48, and Ala86) are not conserved in Mu8.1 (Ala32, Asn47, and Phe49) (S16A Fig). Third, the GluA2-binding surface of con-ikot-ikot is negatively charged (S16B Fig), whereas the corresponding surface of Mu8.1 is predominantly positively charged (S16C Fig). Fourth, although the Mu8.1 dimer surface is negatively charged as in con-ikot-ikot, the 2 dimers are of unequal dimensions (S16D Fig).

The voltage-gated Cav2.3 channel was identified as the highest affinity target of Mu8.1 among an extensive collection of mammalian ion channels and GPCRs. Most of the knowledge about the function of Cav2.3 has been obtained from animal knockout and cellular knockdown experiments, where the channel was linked to epilepsy, neurodegeneration, and pain [50–53]. In contrast to SNX-482, combined results from calcium imaging and electrophysiology measurements suggest that Mu8.1 modulates sensory neurons via inhibition of Cav2.3

channels without evidencing cross-actions against other neuronal conductances typically associated with somatosensory subclasses. Thus, the Mu8.1 scaffold could serve as a valuable addition to the existing molecular toolbox for investigating the physiological functions of Cav2.3, as well as its involvement in synaptic signaling and neuromodulation. Improvements guided by structure–function analysis using, for instance, mutational screening, could further enhance the potency and selectivity of Mu8.1, thereby augmenting its utility as a molecular tool.

At a general level, this study illustrates that a combination of data mining and recombinant expression in *E. coli* can pave the way for a detailed analysis of structural and functional features of newly identified macro-conotoxins. We propose that with the advent of *E. coli* expression systems such as csCyDisCo and DisCoTune [25,28,54], as well as others [55,56], the time is ripe to begin the systematic exploration of a new realm of macro-conotoxins. These efforts will help provide a better understanding of the biological correlates of having large venom components, with the overarching aim of connecting the biochemical and molecular characteristics of venom components with the biology and behavior of cone snails.

## Materials and methods

### Venom gland transcriptome analysis

RNA extraction and transcriptome sequencing and assembly were performed as described previously [57,58]. Assembled transcripts were annotated using a BLASTx search [22] (E-value setting of $1 \times 10^{-3}$) against a combined database derived from UniProt, Conoserver [59], and an in-house cone snail venom transcript library. The 2 toxins, SLC_Mu8.1 and SLC_Mu8.1ii, abbreviated as Mu8.1 and Mu8.1ii, were named according to [21]. Here, Mu describes the 2-letter species abbreviation (Mu for *C. mucronatus*), 8 describes the cysteine scaffold, and the number 1 represents the first toxin to be described from this gene family. The suffix ii is given to a sequence variant that likely represents an allelic variant.

### Transcriptome mining of the NCBI, DDBJ, and CNGB databases

To find sequences that share similarity with Mu8.1, we mined the transcriptomes of 37 cone snail venom gland transcriptomes available in the NCBI, DDBJ, and CNGB repositories using the precursor sequence of Mu8.1 as query (accession numbers provided in S1 Table). Transcriptome assemblies were done as described previously [57,58]. Signal and propeptide sequences of the identified homologous sequences were predicted using ProP v. 1.0 [60]. Mature toxin sequences were predicted to begin after the last basic amino acid residue preceding the first cysteine in the sequence. Therefore, further trimming of sequences to meet this criterion was executed manually as needed. Multiple sequence alignments were carried out using the MAFFT version 7 multiple alignment online interface [61] and visualized in Jalview version 1.0 [62].

### Plasmid generation

The plasmid for bacterial expression of Mu8.1 was generated by uracil excision cloning, as described previously [63]. Polymerase chain reaction was carried out using Phusion U Hot Start polymerase (Thermo Fisher Scientific) according to the manufacturer's instructions. Based on the transcriptome data for Mu8.1, the sequence of the mature toxin was predicted as GENSDNLTHCRLFEFRLCLLECMSLTLDHCYARCTTVITQIHGSDTNRFDCTIFKTCYYR CYVLGKTEDHCWKGTATSVTGDVGDLEFC. A codon-optimized DNA sequence for bacterial expression was generated using the CodonOpt tool. Using this codon-optimized DNA sequence as a template, the following 2 primers were designed:

Mu8.1_sense: ACACGGAUCGGACACCAATCGTTTTGATTGCACAATCTTCAAGAC
CTGCTACTACCGGTGCTACGTTCTTGGTAAAACAGAAGACCATTGCTGGAAAGG
GACGGCAACGTCAGTGACAGGTGATGTCGGAGATTTGGAATTTTGCTAAGAATTC
GAGCTCCGTCGACAG;Mu8.1_antisense:
ATCCGTGUATCTGGGTAATAACTGTGGTACATCTCGCATAGCAGTGGTCTAATG
TAAGCGACATACACTCCAACAAACACAGCCGGAACTCGAATAATCTACAATGGGT
AAGGTTGTCTGAGTTTTCGCCCTGAAAATACAGATTCTCAC.

These primers were subsequently used to clone Mu8.1 into the pET39_Ub19 expression vector [64]. The resulting plasmid encoding Ub-His$_{10}$-tagged Mu8.1 (Ub-His$_{10}$-Mu8.1) is referred to as pLE601. The fusion protein produced from pLE601 also contains a TEV protease recognition site following the Ub–His$_{10}$-tag. Primers were purchased from Integrated DNA Technologies, and the sequence encoding Ub-His$_{10}$-Mu8.1 was confirmed by Eurofins.

## Protein expression

Chemically competent *E. coli* BL21 (DE3) cells were transformed with pLE601 with or (as a control) without the csCyDisCo plasmid (pLE577) [25]. Cells were plated on lysogeny broth (LB) agar supplemented with kanamycin (50 μg/mL) with (when cotransforming with pLE577) or without chloramphenicol (30 μg/mL). A single colony was picked to inoculate the LB medium containing the same type and concentration of antibiotic as used on the LB agar plates. The overnight culture was incubated for approximately 16 hours at 37°C at 200 rpm in an orbital shaker.

For initial small-scale expression tests (50 mL), LB medium containing appropriate antibiotics and supplemented with 0.05% glucose was inoculated with 2% overnight culture and grown at 37°C with shaking at 200 rpm until the desired OD$_{600}$ of 0.6 to 0.8 was reached. Expression was induced by adding isopropyl ß-D-1-thiogalactopyranoside (IPTG) to a final concentration of 1 mM and the cultures grown for 18 hours at 25°C with shaking at 200 rpm to allow protein expression. Large-scale expression (1 L culture volume) was performed in autoinduction media prepared as described previously [65]. Briefly, terrific broth medium containing kanamycin (100 μg/mL) and/or chloramphenicol (30 μg/mL) was supplemented with sterilized stocks of the following: 0.05% glucose, 0.2% lactose, 50 mM KH$_2$PO$_4$/Na$_2$HPO$_4$, 50 mM NH$_4$Cl, 50 mM Na$_2$SO$_4$, 0.1 mM FeCl$_3$, 2 mM MgSO$_4$, 0.1 mM CaCl$_2$, and 1 × metal mix (203 g/L MgCl$_2$ 6·H$_2$O, 2.1 g/L CaCl$_2$ 2·H$_2$O, 2.7 g/L FeSO$_4$ 7·H$_2$O, 20 mg/L AlCl$_3$ 6·H$_2$O, 10 mg/L CoSO$_4$ 7·H$_2$O, 2 mg/L KCr(SO$_4$)$_2$ 12·H$_2$O, 2 mg/L CuCl$_2$ 2·H$_2$O, 1 mg/L H$_3$BO$_4$, 20 mg/L KI, 20 mg/L MnSO$_4$ H$_2$O, 1 mg/L NiSO$_4$ 6·H$_2$O, 4 mg/L Na$_2$ MoO$_4$ 2·H$_2$O, 4 mg/L ZnSO$_4$ 7·H$_2$O, 21 g/L citric acid monohydrate). Cultures were grown at 37°C at 200 rpm until OD$_{600}$ reached 0.8, at which point cells were moved to 25°C for expression performed with shaking at 200 rpm for 18 hours.

## Harvest and clarification of bacterial cultures

Induced cultures were harvested by centrifugation at 5,000*g* for 20 minutes. The cell pellets were resuspended in 5 mL lysis buffer (50 mM Tris (pH 8), 300 mM NaCl, 20 mM imidazole) per gram pellet. Cell resuspensions were supplemented with approximately 12 units Benzonase Nuclease (Merck Millipore)/L culture to minimize viscosity due to the presence of nucleic acids post-lysis. Cell lysis was performed using a UP200S ultrasonic processor (Hielscher) keeping the cells on ice throughout. Cells were lysed with 8 × 30-second pulses at 90% power with 30-second rests between each pulse. Cell debris was pelleted by centrifugation at 30,000*g* for 45 minutes. The cleared lysates were filtered through 0.45 μm syringe filters and transferred

to fresh tubes, whereas the pellets were resuspended in an equal volume lysis buffer containing 8 M urea for SDS-PAGE analysis.

## Protein purification

Ub-His$_{10}$-Mu8.1 was affinity purified from the clarified lysate on an ÄKTA START system equipped with a 5-mL prepacked HisTrap HP (Cytiva) column equilibrated in lysis buffer. The lysate was applied to the column and washed with approximately 20 column volumes (CVs) of lysis buffer before elution of Ub-His$_{10}$-Mu8.1 with a gradient of 0% to 100% elution buffer (50 mM Tris (pH 8), 300 mM NaCl, 400 mM imidazole) applied over 20 CVs. Pooled fractions were dialyzed twice against 2 L anion exchange (AEX) buffer (50 mM NaH$_2$PO$_4$/Na$_2$HPO$_4$ (pH 6.8), 20 mM NaCl). AEX chromatography was performed on an ÄKTA Pure system equipped with a 10/100 Tricorn column (Cytiva) packed with Source 15Q ion exchange resin (Amersham Biosciences, GE Healthcare) equilibrated in AEX buffer. Ub-His$_{10}$-Mu8.1 was eluted using a gradient from 15% to 50% AEX elution buffer (50 mM NaH$_2$PO$_4$/Na$_2$HPO$_4$ (pH 6.8), 1 M NaCl) developed over 6 CVs.

The Ub-His$_{10}$-Mu8.1 fusion protein was cleaved using His$_6$-tagged TEV protease, expressed, and purified as described previously [25]. A molar ratio Ub-His$_{10}$-Mu8.1:His$_6$-TEV protease of 1:20 was used. To avoid reducing the disulfides in Mu8.1, His$_6$-TEV protease—pre-activated with 2 mM dithiothreitol (DTT) for 30 minutes at room temperature—was diluted to approximately 0.002 mM DTT by 3 rounds of dilution/concentration in an Amicon Ultra 15 mL 3K Centrifugal Filter (Merck Millipore). TEV protease cleavage was performed overnight at room temperature.

To remove uncleaved Ub-His$_{10}$-Mu8.1, the Ub-His$_{10}$ tag, and His$_6$-TEV, the cleavage mixture was applied to a gravity flow column packed with 8 mL TALON cobalt resin (Takara) equilibrated in AEX buffer. The flow-through and the first wash fraction were collected. The presence of cleaved Mu8.1 in each fraction was investigated by analysis on 15% tricine SDS-PAGE gels, and the protein-containing fractions pooled. The cleaved Mu8.1 was subjected to size exclusion chromatography on a Superdex 75 Increase 10/300 GL column (Cytiva) equilibrated in 200 mM NH$_4$HCO$_3$ buffer (pH 7.8). Fractions containing purified Mu8.1 were pooled and lyophilized.

## Analytical gel filtration

To analyze the oligomeric state of Mu8.1, the protein was analyzed at 2 concentrations, 100 μm and 1 μm, on a Superdex 75 Increase 10/300 GL column (Cytiva) equilibrated in 10 mM NaPi (pH 7.8), 150 mM NaCl at a flowrate of 0.5 mL/min. An excess of the protein was loaded onto a 100 μL loop to ensure analysis of the same volume in each run.

## SDS-PAGE analysis

Samples from bacterial expression and subsequent purification steps were separated on 15% glycine SDS-PAGE or 15% tricine SDS-PAGE gels [66]. Where indicated, reduced samples were treated with 40 mM DTT. Protein bands were visualized with Coomassie Brilliant Blue, and images were recorded with a BioRad Chemidoc Imaging System. For western blotting, proteins separated by SDS-PAGE were transferred to a PVDF membrane (Immobilon-P, Merck Millipore) in a Mini Trans-Blot (Bio-Rad) transfer system. A mouse monoclonal His-tetra (Qiagen) antibody (1:1,000 dilution) was used in combination with a horseradish peroxidase–conjugated α-mouse secondary antibody (Pierce) (1:100,000 dilution). Chemiluminescence detection was performed using ECL Select Peroxide and Luminol solutions (GE Healthcare) according to the manufacturer's directions.

## Concentration determination

Concentrations were determined by measuring absorbance at 280 nm and using the theoretical extinction coefficient provided by the Expasy ProtParam tool available through the Expasy bioinformatics resource web portal [67]. Concentrations used for bioassays, constellation pharmacology, and electrophysiology assumed a monomeric state of Mu8.1.

## Determination of molecular mass in solution by small angle X-ray scattering (SAXS)

Prior to SAXS analysis, Mu8.1 was run through a Superdex 75 Increase 10/300 GL column to ensure a monodisperse sample. The protein was subsequently dialyzed into a solution containing 150 mM NaCl and 10 mM NaPi (pH 7.8), and the dialysis buffer was reserved for measurement of background scattering. Immediately preceding SAXS measurements any precipitates were removed from the sample by centrifugation at 20,000$g$ for 15 minutes at 4˚C. Six dilutions were prepared ranging from 0.5 mg/mL to 12.4 mg/mL (49 μM to 1.2 mM) in dilution buffer. SAXS data were collected by the beamline staff at CPHSAXS using a Xenocs BioXolver L equipped with a liquid gallium X-ray source ($\lambda$ = 1.34 Å). A sample-to-detector distance of 632.5 mm was used, corresponding to a Q-range of 0.013 to 0.5 Å$^{-1}$. Samples and buffer were measured at room temperature, and automatic loading was performed robotically from a 96-well plate. Data were collected as a minimum of 10 frames with 120-second exposure per frame. The longer exposure time was to account for the lower concentration and the expected presence of multimeric species. Data reduction and primary analysis were performed using RAW [68], singular value decomposition (SVD), and oligomer analysis performed with OLIGOMER (ATSAS program package) [69], and scattering curves plotted in Matlab R2020b.

## X-ray crystallography

Freeze-dried Mu8.1 was dissolved in Milli-Q water to a concentration of 5 mg/mL. Crystallization screening experiments were performed with the Structure screen II (Molecular Dimensions, MD1-02) and the Index screen (Hampton Research, HR2-144) by the hanging drop vapor diffusion method. The crystal drops were mixed using 1 μL of protein and 1 μL precipitant solution as hanging drops on siliconized glass cover slides and equilibrated against 500 μL of precipitant solution in a 24-well plate setup. Wells were sealed with immersion oil (Sigma-Aldrich) and incubated at 21˚C. Initial crystals of Mu8.1 appeared in several conditions in a few days to weeks. Diffracting crystals were obtained from the Structure screen II condition 38 (0.1 M NaOAc (pH 4.6), 0.1 M CdCl hemi(pentahydrate), 30% v/v PEG400), abbreviated as Mu8.1_38, and from the Index screen condition 59 (0.1 M HEPES (pH 7.5), 0.02 M magnesium chloride hexahydrate, 22% w/v polyacrylic acid sodium salt 5,100), abbreviated as Mu8.1_59.

Crystals were harvested using mounted CryoLoops (Hampton Research) and flash-cooled in liquid nitrogen. Cryoprotection was performed by quickly dipping the crystal in approximately 17% ethylene glycol (EG) prepared by mixing 1 μL 50% EG with 2 μL of the reservoir condition specific to each crystal condition. Both native crystals and iodide-soaked crystals were prepared from all 3 conditions. Single iodide crystals (Sigma-Aldrich) were added to the abovementioned cryo condition for each of the crystal conditions. Crystals from each condition were transferred to the iodide-containing cryo conditions and left to soak 5 to 10 seconds before harvesting them.

## Data collection

Flash-cooled crystals were shipped to the beamline for remote data collection. Data were collected at 100K on a PILATUS detector at BioMax (MAX-IV, Lund, Sweden). A full sweep of 360˚ data was collected with an oscillation degree of 0.1˚, with 0.050-second exposure at 12,700 eV and 7,000 eV. Complete data set was processed from 360˚ (3,600 images) with the X-ray beam reduced to 5% intensity.

## Data processing

Native data were collected for Mu8.1_59 at 12,700 eV, and only Mu8.1_38 showed an anomalous signal from the data collected at 7,000 eV. All data were processed with xia2 using the 3dii pipeline [70,71] (Table 1).

The phases for Mu8.1_38 were experimentally determined using autosol in the PHENIX package [72] with 13 iodide sites identified and an initial figure of merit (FOM) of 0.4. Density resembling helical structures were visible in the electron density map. The following AutoBuild wizard within the PHENIX package [72] was able to build a preliminary structure with the main helices in place. This structure was used as initial search model for molecular replacement and performed with the program Phaser [73] against the highest-resolution dataset Mu8.1_59.

All structures were manually refined using phenix.refine [72], and final model building was performed in Coot [74]. Data collection and refinement statistics are summarized in Table 1. Molecular graphics were presented with the PyMOL Molecular Graphics System, Version 2.2r7pre, Schrödinger, LLC. Electrostatic potentials were modelled using the Adaptive Poisson-Boltzmann Solver (APBS) plugin in PyMOL2.3 [75].

## Structure search and topological comparisons

Structures similar to Mu8.1 were identified using PDBeFold at the European Bioinformatics Institute (https://www.ebi.ac.uk/msd-srv/ssm/) [76,77]. Monomeric Mu8.1 was used as query molecule to identify similar structures with a minimum acceptable match set to 60% or higher from the entire PDB database. Structural overlays were generated with the CLICK structural alignment tool (http://cospi.iiserpune.ac.in/click/) selecting "CA" as representative atoms and visualized using the PyMOL Molecular Graphics System [78] or executed manually in PyMol.

## Constellation pharmacology

Primary cell cultures were dissociated from CGRP-GFP mice, STOCK Tg(Calca-EGFP) FG104Gsat/Mmucd, ages 34 to 38 days old as described previously [11]. In brief, lumbar DRG from vertebrae L1 to L6 were dissected, trimmed, and treated with 0.25% trypsin for 20 minutes. Following trypsinization, the DRGs were mechanically triturated using fire polished pipettes and plated in poly-l-lysine–coated plates. All plated cells were kept overnight at 37˚C in a minimal essential medium supplemented with 10% fetal bovine serum, 1X penicillin/streptomycin, 10 mM HEPES, and 0.4% (w/v) glucose. One hour before the experiment, the dissociated cells were loaded with 4 μM Fura-2-AM dye (Sigma-Aldrich) and kept at 37˚C. During each experiment, all dissociated cells were exposed to different pharmacological agents utilizing an automatic perfusion system and were imaged at 340/380 nm at 2 frames per second. In brief, cells were incubated with the pharmacological agents for 15 seconds followed by 6 consecutive washes and a 5-minute incubation period with extracellular solution for controls or Mu8.1. Five different pharmacological agents were used for cell classification: mustard oil (AITC) at 100 μM, menthol at 400 μM, capsaicin at 300 nM, $K^+$ at 25 and 40 mM, and

**Table 1. Data collection and refinement statistics.**

| Name (PDB ID) | Mu8.1_59* (7PX2) | Mu8.1_38* (7PX1) |
|---|---|---|
| Wavelength (Å) | 1.771 | 1.771 |
| Resolution range (Å) | 48.97–2.12 (2.196–2.12) | 36.62–2.33 (2.413–2.33) |
| Space group | P 1 21 1 | P 21 21 2 |
| a, b, c (Å) | 68.8, 88.4, 69.3 | 49.3, 109.3, 34.3 |
| α, β, γ (°) | 90, 116.7, 90 | 90, 90, 90 |
| Total reflections | 258,079 (14,423) | 103,396 (10,360) |
| Unique reflections | 41,056 (3,498) | 8,423 (811) |
| Multiplicity | 6.3 (4.1) | 12.3 (12.8) |
| Completeness (%) | 97.3 (83.2) | 99.7 (99.9) |
| Mean I/sigma (I) | 9.75 (1.21) | 7.62 (1.08) |
| Wilson B-factor(Å$^2$) | 43.38 | 47.60 |
| R-merge | 0.091 (0.81) | 0.21 (>1) |
| R-meas | 0.099 (0.92) | 0.21 (>1) |
| R-pim | 0.039 (0.44) | 0.064 (0.38) |
| CC1/2 | 0.99 (0.68) | 0.99 (0.60) |
| CC* | 0.99 (0.89) | 0.99 (0.87) |
| Reflections used in refinement | 41,043 (3,498) | 8,408 (812) |
| Reflections used for R-free | 2,082 (167) | 446 (42) |
| R-work | 0.21 (0.38) | 0.26 (0.37) |
| R-free | 0.23 (0.38) | 0.30 (0.48) |
| CC (work) | 0.95 (0.80) | 0.92 (0.81) |
| CC (free) | 0.96 (0.80) | 0.87 (0.57) |
| Number of non-hydrogen atoms | 4,299 | 1,386 |
| Macromolecules | 4,086 | 1,346 |
| Ligands | 192 | 13 |
| Solvent | 123 | 27 |
| Protein residues | 510 | 168 |
| RMSD (bonds) (Å) | 0.006 | 0.003 |
| RMSD (angles) (°) | 1.03 | 0.49 |
| Ramachandran favored (%) | 99.00 | 97.56 |
| Ramachandran allowed (%) | 1.00 | 1.83 |
| Ramachandran outliers (%) | 0.00 | 0.61 |
| Rotamer outliers (%) | 0.64 | 0.65 |
| Clash score | 5.81 | 5.72 |
| Average B-factor (Å$^2$) | 58.2 | 53.6 |
| Macromolecules | 58.00 | 53.40 |
| Ligands | 64.69 | 78.83 |
| Solvent | 60.20 | 54.09 |
| Number of TLS groups | 18 | 10 |

*Statistics for the highest-resolution shell are shown in parentheses.

conotoxin κM-RIIIJ at 1 μM as described previously [11]. SNX-482 (Alomone Labs) was used at 100 nM. At the end of each experiment, all cells were incubated for 7 minutes with a 2.5-μg/mL Alexa Fluor 647 Azolectin B4 (IB4). All data acquisition was performed using the Nikon NIS-Elements platform. CellProfiler [79] was used for region of interest (ROI) selection, and a custom-built script in python and R was used for further data analysis and visualization. The

package used to visualize, analyze, and deploy models for constellation pharmacology experiments is available at https://github.com/leeleavitt/procPharm. All procedures were approved by the University of Utah institutional animal care and use committee (IUCAC) Protocol number: 17–05017.

## Statistical analysis of cellular calcium imaging

We utilized a min-max normalization (f(x)) to assess the effects of Mu8.1 on the peak height of the calcium signal induced by high concentrations of potassium using the formula:

$$f(x) = \frac{K^+_{test} - K^+_{control}}{K^+_{test} + K^{+.}_{control}}$$

where $K^+$ test is the peak height after the incubation with Mu8.1, and $K^+$ control is the peak height before the incubation with Mu8.1 [11]. If f(x) = 0, this suggests that the conotoxin did not affect the $Ca^{2+}$ concentration in the cytosol induced by the high concentration of potassium. An f(x) > 0 indicates that the conotoxin increased the cytosolic calcium concentration after a high potassium concentration, resulting in amplification. Finally, if f(x) < 0, the conotoxin decreased calcium concentration in the cytosol resulting in a calcium block. After calculating the f(x) for all sensory neuron cell types, we performed a two-tailed $t$ test to assess if the f(x) values calculated for every cell type were significantly different from 0.

## Electrophysiology

APC recordings were performed in a Patchliner Octo (Nanion Technologies GmbH, Munich, Germany) equipped with 2 EPC-10 quadro patch clamp amplifiers (HEKA Electronics). PatchControlHT (Nanion) was used for cell capture, seal formation, and establishment of the whole-cell configuration, while voltage was controlled, and currents sampled with PatchMaster (HEKA Electronik). Recordings were performed under the whole-cell configuration using single-hole planar NPC-16 chips (resistance of approximately 2.5 MΩ) at room temperature. Stably transfected cell lines (D.J. Adams collection, IHMRI-UOW) were cultured according to the manufacturer's instructions and detached using TrypLE. Cells were resuspended in cold external recording solution and kept in suspension by automatic pipetting at 4°C. The extracellular solution used for Kv1, Kv4.3, hERG, and Nav1 recordings contained (in mM): 140 NaCl, 5 KCl, 2 CaCl$_2$, 2 MgCl$_2$, 10 glucose, and 10 HEPES (pH 7.4 with NaOH, 298 mOsmol/kg). Kv1, Kv4.3 and hERG intracellular solution (in mM): 60 KF, 70 KCl, 10 EGTA, 10 glucose, and 10 HEPES (pH 7.2 with KOH, 285 mOsmol/kg). Peak $I_K$ currents for Kv1.1–3 and Kv4.3: 500 ms test pulse to 20 mV (Vh = −120 mV; 0.1 Hz). hERG $I_K$: 1-second prepulse to +40 mV was followed by 200 ms test pulse to −40 mV (Vh = −80 mV; 0.1 Hz). Nav1 intracellular solution (mM): 60 CsF, 60 CsCl, 10 NaCl, 10 EGTA, and 10 HEPES (pH 7.2 with CsOH, 285 mOsmol/kg). Nav currents were elicited by 10 ms test pulses to −10 mV with a 1-second prepulse to −120 mV (Vh = −90 mV; 0.1 Hz). APC of Cav1 and Cav2: extracellular solution (in mM): 135 NaCl, 4 KCl, 10 BaCl$_2$, 1 MgCl$_2$, 5 glucose, and 10 HEPES (pH 7.4 with NaOH, 298 mOsmol/kg) and intracellular solution (in mM): 90 CsSO$_4$, 10 CsCl, 10 NaCl, 10 EGTA, and 10 HEPES (pH 7.2 with CsOH, 285 mOsmol/kg). Peak calcium currents were measured upon 50 ms step depolarization to +10 mV (Vh = −80 mV; 0.1 Hz). Recordings where seal resistance (SR) was >500 MΩ and access resistance was <3xSR were considered acceptable. Chip and whole-cell capacitance were fully compensated, and series resistance compensation (70%) was applied via Auto Rs Comp function. Recordings were acquired with PatchMaster (HEKA Elektronik, Lambrecht/Pfalz, Germany) and stored on a computer running PatchControlHT software (Nanion Technologies GmbH, Munich, Germany).

Manual patch clamp (MPC) was performed on human embryonic kidney (HEK293T) cells containing the SV40 Large T-antigen cultured and transiently transfected by calcium phosphate method as reported previously [80]. In brief, cells were cultured at 37˚C, 5% $CO_2$ in Dulbecco's Modified Eagle's Medium (DMEM, Invitrogen Life Technologies, Victoria, Australia), supplemented with 10% fetal bovine serum (Bovigen, Victoria, Australia), 1% GlutaMAX and penicillin–streptomycin (Invitrogen). The human orthologues of Cav3.1, Cav3.2, and Cav3.3 channels were cotransfected with GFP for identification of positive transfectants. cDNAs encoding hCav3.1 (kindly provided by G. Zamponi, University of Calgary), hCav3.2 (a1Ha-pcDNA3, Addgene #45809), and hCav3.3 (a1Ic-HE3-pcDNA3, Addgene #45810) were a kind gift from E. Perez-Reyes (University of Virginia). MPC experiments employed a MultiClamp 700B amplifier, digitalized with a DigiData1440, and controlled using Clampex11.1 software (Molecular Devices, California, USA). Recordings of $I_{Ca}$ through hCav3.1–3 were performed using an extracellular solution containing (in mM): 110 NaCl, 10 CaCl$_2$, 1 MgCl$_2$, 5 CsCl, 30 TEA-Cl, 10 D-Glucose, and 10 HEPES (pH 7.35 with TEA-OH, 305 mOsmol/kg). Pipettes were pulled from borosilicate glass capillaries (GC150F-15, Harvard Apparatus, Massachusetts, USA), fire polished to a final resistance of 1 to 3 MΩ, and filled with intracellular solution (in mM): 140 KGluconate, 5 NaCl, 2 MgCl$_2$, 5 EGTA, and 10 HEPES (pH 7.2 with KOH, 295 mOsmol/kg). Peak currents were measured upon stimulation using 50 ms test pulses to −20 mV from a holding potential (Vh) of −90 mV and pulsed at 0.2 Hz. Whole-cell currents were sampled at 100 kHz and filtered to 10 kHz, with leak and capacitive currents subtracted using a P/4 protocol, and 60% to 80% series resistance compensation.

## Data analysis of electrophysiology experiments

APC analysis was performed using Igor Pro-6.37 (WaveMetrics). Cav2.3 peak currents measured in the presence of increasing Mu8.1 concentrations ($I_{Mu8.1}$) were divided by the current in control conditions ($I_{Ctr}$) to generate a concentration–response curve that was fit with a Hill equation of the form:

$$\left(1 - \frac{I_{Mu8.1}}{I_{Ctr}}\right) = \frac{IC50^h}{IC50^h + [Mu8.1]^h}$$

where $IC_{50}$ is the half-maximal inhibitory concentration, and h is the Hill coefficient (nH).

For ease of comparison, $IC_{50}$ values were calculated from fractional inhibition for the other voltage-gated channels that were screened at a single Mu8.1 concentration according to the following equation:

$$IC_{50} = \frac{I_{Mu8.1}}{1 - I_{Mu8.1}}[Mu8.1].$$

## Supporting information

**S1 Fig. Mu8.1 represents a new class of conotoxins.** Sequences harvested from venom gland transcriptomes available in the NCBI, DDBJ, and CNGB repositories using the precursor sequence of Mu8.1 as query. Truncated sequences, duplicates, and variants with only 1–2 amino acid residue differences were not included. The remaining sequences were used in a multiple sequence alignment carried out using the MAFFT version 7 multiple alignment online interface [61] and visualized in Jalview [62]. For clarity, the signal and propeptide sequences are depicted with a space preceding the mature toxin sequences. Amino acid residues are shaded in gray according to a 90% identity threshold (all cysteine residues are shaded yellow regardless of conservation). Prepro-sequences are annotated with colored bars

(positioned according to the Mu8.1 sequence) indicating the tripartite organization. Mauve: signal sequence; maroon: propeptide region; green: mature conotoxin, here drawn to illustrate the α-helical structure as it corresponds to the Mu8.1 sequence. Red asterisks indicate those sequences previously annotated as con-ikot-ikot uncovered by pBLAST searching as described in the main text. Red arrows highlight amino acid residues referred to throughout the main text, and green rectangles represent α-helices.
(TIF)

**S2 Fig. BLOSUM62 cluster map of con-ikot-ikot–related conotoxin precursors.** The nodes depict individual precursor amino acid sequences, and the edges correspond to the BLAST $p$-values $< 1 \times 10^{-6}$ between the nodes. The 4 clusters are labelled and color-coded according to the legend in the upper, right-hand corner. Mu8.1 and Mu8.1ii are furthermore highlighted by the star shapes. Processed data for cluster analysis found in S1 Data.
(TIF)

**S3 Fig. Multiple sequence alignment of Mu8.1 and randomly selected sequences representing the 4 toxin clusters.** Only the 5′ UTRs and the beginning of ORFs are shown. The start codon is shown in yellow, and columns with $\geq 75\%$ sequence identity are highlighted in black. The sequence names are colored to match the clusters in S2 Fig.
(TIF)

**S4 Fig. Gene structure of *C. ventricosus* and *C. betulinus* conotoxins from the 4 clusters in S2 Fig.** Two transcripts from *C. ventricosus* and 6 from *C. betulinus* were identified that could be successfully mapped to the respective genomes. These transcripts were used to assess intron locations and phases from each of the 4 clusters. The exons are represented by wide boxes proportional to the length of the sequences, whereas the introns are shown by thin interspaced segments (not proportional to sequence length) with their phases given above each intron. The predicted signal sequence is colored grey, and the remaining precursor colored to match S2 Fig.
(TIF)

**S5 Fig. (A) Recombinantly expressed Mu8.1 purifies as a single, fully oxidized species from *E. coli*.** Schematic overview of the 2 plasmids of the csCyDisCo expression system [25] used for recombinant expression of Ub-His$_{10}$-Mu8.1. The csCyDisCo plasmid (pLE577) encodes for the 3 enzymes, Erv1p, hPDI, and csPDI. pLE601 encodes the Ub-His$_{10}$-Mu8.1 fusion protein as shown in the light gray box. (**B**) MALDI-TOF spectrum of nonreduced Mu8.1 showing a single peak with a mass of 10,181.7 Da. The theoretical average mass of Mu8.1 is 10,181.5 Da. (**C**) CD spectrum of nonreduced Mu8.1 recorded at 25°C. The blue line represents an average of 10 scans. Blue shading around the curve signifies the standard error of the mean of the 10 recorded spectra. CD, circular dichroism; csPDI, conotoxin-specific PDI; hPDI, human PDI; MALDI-TOF, matrix-assisted laser desorption–ionization time of flight; Ub-His$_{10}$, Ub containing 10 consecutive histidines.
(TIF)

**S6 Fig. Mu8.1 exists primarily as a dimer in solution. (A)** SAXS scattering profiles of increasing concentrations of Mu8.1 dissolved in 10 mM NaPi (pH 8), 150 NaCl. Inset: Guinier plots of scattering profiles, where straight lines were obtained by linear regression of the scattering profiles in the low q2 region. The Guinier region is highlighted with circles. Source data for quantifications provided in S2 Data.
(TIF)

**S7 Fig. Molecules of the asymmetric unit of Mu8.1_59.** The asymmetric unit of Mu8.1_59 accommodates 6 molecules that form 3 equivalent dimers. The dimers are shown in orange, green, and dark purple.
(TIF)

**S8 Fig. Mu8.1 does not inhibit desensitization of the GluA2 AMPA receptor.** Intracellular $Ca^{2+}$ imaging was used to determine a possible effect of Mu8.1 on the AMPA receptor GluA2 as described in Materials and methods. The experiment was performed in the absence (left) and presence (right) of cyclothiazide (CTZ), a positive allosteric modulator of AMPA receptors, known to block receptor desensitization and thus increase AMPA receptor current. Black arrows indicate the addition of saturating agonist solution (1 mM glutamate). The experiment was conducted at different concentrations of Mu8.1 as indicated. No effect of Mu8.1 treatment was observed. Source data provided in S5 Data.
(TIF)

**S9 Fig. Current–voltage relationship of Cav2.3-mediated currents in control and the presence of Mu8.1. (A)** Representative family of Cav2.3 currents elicited by a standard IV protocol (50 ms pulses from −80 mV to 60 mV in 10 mV steps, Vh −90 mV) in the absence (control, green) and the presence of Mu8.1 (10 μM, mauve). Scale bar is 2 nA. (**B**) Average I-V plots from peak currents in control ($' V_{0.5} = -22.1 \pm 2.4$ mV, $n = 4$) and Mu8.1 ($' 3$ μM $V_{0.5} = -18.5 \pm 1.9$, $n = 4$; and $' 10$ μM $V_{0.5} = -18.0 \pm 2.5$, $n = 4$). One-way ANOVA with Dunnett multiple comparisons test of control vs. 3 μM Mu3.1 $p = 0.4402$; control vs. 10 μM $p = 0.3587$. Source data and quantifications provided in S6 Data.
(TIF)

**S10 Fig. Time course of Mu8.1 inhibition and recovery of Cav2.3-mediated current. (A)** Representative APC whole-cell peak currents recorded from a cell expressing Cav2.3 channels during application of Mu8.1 (10 μM, shaded box) and upon full extracellular bath solution washout. Stimuli: 50 ms pulses to −20 mV at 0.2 Hz (Vh −90 mV). Source data provided in S7 Data.
(TIF)

**S11 Fig. Receptor and GPCRome screening of Mu8.1. (A)** Primary radioligand binding assay of Mu8.1 (10 uM) at 52 receptors and ion channels (x-axis). Plotted values represent means ($n = 4$), and error bars are ±SEM. Arrows show 2 receptors, 5-HT3 and histamine H1, with inhibition >50% that were selected for secondary testing. Source data and quantifications provided in S8 Data. (**B**) Secondary binding assay of the 5-HT3 (left) and H1 (right) receptors. Plotted values represent means ($n = 3$), and error bars are ±SEM. The results demonstrate that the 2 initial hits from S11A were false positives. (**C**) PRESTO-Tango screen of Mu8.1 (10 μM) at 318 human GPCRs (x-axis). Plotted values represent means ±SD. The arrow indicates a potential hit (ADRAC2) that met the cutoff for further testing. However, a closer examination of the data showed a false positive due to an outlier.
(TIF)

**S12 Fig. Mu8.1 irreversibly inhibits $Ca^{2+}$ influx in a subpopulation of peptidergic nociceptors. (A)** Examples of calcium traces from peptidergic nociceptors in which Mu8.1 inhibition of the peak was not reversed upon washout. Treatment with 10 μM Mu8.1 was scored as "irreversible" if the second $Ca^{2+}$ peak after Mu8.1 treatment was lower than the peak immediately before Mu8.1 treatment (see mauve dotted line). Each trace represents the calcium signal (ΔF/F) of the neuron pictured on the left (GFP-CGRP+: peptidergic nociceptors; Alexa Fluor 647-Isolectin B4+: nonpeptidergic nociceptors, and bright-field). KCl depolarization pulses

(25 mM) are indicated by light grey shading. A higher KCl pulse (40 mM) was used to elicit a maximum calcium signal at the end of the experiment. Class-defining pharmacology: RIIIJ (1 μM, blue box); AITC (100 μM; mustard flower), menthol (400 μM; peppermint leaf), and capsaicin (300 nM; chili pepper). Horizontal lines flanking breaks within a trace signify graphical adjustment of trace amplitude to avoid overlap of neighboring traces. (**B**) Peptidergic nociceptor populations from 3 independent experiments showing the percentage of neurons (within the bars) where Mu8.1 treatment was reversible (empty) or deemed irreversible (filled). The number of peptidergic nociceptive neurons recorded in each experiment is shown in parentheses. Source data of individual traces and for quantifications shown in S9 Data. AITC, allyl isothiocyanate; CGRP, calcitonin gene-related peptide; GFP, green fluorescent protein. (TIF)

**S13 Fig. SNX-482 amplifies intracellular Ca$^{2+}$ signal in subpopulations of sensory neurons.** Examples of calcium traces from peptidergic nociceptors and C-LTMRs in which SNX-482 amplified the KCl depolarization-induced Ca$^{2+}$ peak (see pink dotted line). Each trace represents the calcium signal (ΔF/F) of the neuron pictured on the left (GFP-CGRP+: peptidergic nociceptors; Alexa Fluor 647-Isolectin B4+: nonpeptidergic nociceptors, and bright-field). KCl depolarization pulses (25 mM) are indicated by light grey shading. A higher KCl pulse (40 mM) was used to elicit a maximum calcium signal at the end of the experiment. Class-defining pharmacology: RIIIJ (1 μM, blue box); AITC (100 μM; mustard flower), menthol (400 μM; peppermint leaf), and capsaicin (300 nM; chili pepper). Horizontal lines flanking breaks within a trace signify graphical adjustment of trace amplitude to avoid overlap of neighboring traces. (**B**) Neurons affected by SNX-482 from 3 independent experiments showing the percentage of neurons that were amplified or inhibited by SNX-482. The number of neurons in each experiment is shown in parentheses. Source data of individual traces and quantification provided in S10 Data. AITC, allyl isothiocyanate; C-LTMR, C-low threshold mechanoreceptor; CGRP, calcitonin gene-related peptide; GFP, green fluorescent protein. (TIF)

**S14 Fig. Endogenous saposin domain-containing proteins do not share gene structure or sequence similarity with Mu8.1. (A)** Gene structure of saposin domain-containing proteins in *C. ventricosus*. The exons are represented by wide boxes proportional to the length of the sequences, whereas the introns are shown by thin interspaced segments (not proportional to sequence length) with their phases given above each intron. (**B**) Sequence alignment of the saposin domains from Mu8.1 and *C. ventricosus* mesencephalic astrocyte-derived neurotrophic factor (MANF) revealing only little sequence similarity. Source data provided Supplementary_file_D.saposins in S1 Data. (TIF)

**S15 Fig. Toxins from Cluster 1 likely contain a saposin fold. (A)** Graphic representation of AlphaFold-predicted secondary structure elements and disulfide bonds of a Cluster 1 conotoxin from *C. litteratus* (sequence found in Supplementary file A in S1 Data and [81]) (purple) compared to Mu8.1 (green). Disulfide bonds are represented by brackets, and α- and 3$_{10}$-helices are represented as cylinders (numbered by Arabic numerals). The dotted pale green bracket represents a putative disulfide bond not predicted by AlphaFold. (**B**) Cartoon representation of the structure predicted byAlphaFold for the Cluster 1 toxin. The predicted structure displays an α-helical protein with 3 leaf-like domains each consisting of a helix–turn–helix motif. Disulfide bonds are represented by yellow sticks, and free cysteines are represented by pale green sticks. (**C**) The 3 helix–turn–helix motifs individually overlaid with the Mu8.1 crystal structure. Note the close structural similarity (and disulfide pattern) between the first 4

helices of the Cluster 1 protein and Mu8.1, apart from a predicted different orientation between the 2 helix–turn–helix motifs.
(TIF)

**S16 Fig. Structure and sequence features explain Mu8.1's lack of activity at AMPA Receptor GluA2. (A)** The GluA2 AMPA receptor-binding surface of con-ikot-ikot with residues important for binding [29] shown as stick models (left) and the corresponding surface and residues in the Mu8.1 protomer (right). (**B**) Electrostatic surface representation (red: negative; blue: positive) of the con-ikot-ikot GluA2 AMPA receptor-binding surface [29]. (**C**) Electrostatic surface representation of the Mu8.1 protomer—left: the outer surface of the dimer; right: the surface of the dimer interface. (**D**) Electrostatic surface representation of the Mu8.1 dimer showing a negatively charged patch (left). The cartoon representations next to each surface representation indicate orientation.
(TIF)

**S1 Table. Available venom gland transcriptome datasets searched for Mu8.1- and con-ikot-ikot-like sequences.**
(PDF)

**S2 Table. (A) Data reduction and primary analysis of SAXS data.** Data extraction was performed using RAW. Rg Guinier; radius of gyration (Rg) calculated from the slope of the Guinier plot (Panel A, inset); Rg P(r), radius of gyration obtained from pairwise distribution function; I(0)guinier, intensity at zero scattering angle extrapolated from Guinier plot; I(0) P(r), intensity at zero scattering obtained from pairwise distribution function; Dmax, maximum dimension; Mw Bayes, molecular weight from Bayesian inference; Mw Vc, molecular weight from volume of correlation; Primus scale to 1 mg/mL sample, scaling obtained using the Primus program from the ATSAS package [82]; Est concentration from scale, estimated concentration from Primus scale in mg/ml; Mw from scaled concentration (conc), molecular weight as a function of concentration. Source data for quantifications provided in S2 Data. **(B) Percentages of oligomeric species in solution at different concentrations of Mu8.1.** The percentages of oligomeric species from monomer to octamer were determined with Oligomer [69] from the ATSAS package [82]. Source data for quantifications provided in S2 Data.
(PDF)

**S3 Table. Summary of activity determination of Mu8.1 against voltage-gated ion channels.** Automated patch clamp single concentration Mu8.1 screening of recombinantly expressed ion channels. Data for Cav2.3 are provided in Fig 7. Mu8.1 concentrations, test pulse, and $IC_{50}^*$ estimates from single concentration experiments for each channel tested. $IC_{50}^* = \{fc/(1-fc)\} \times [Mu8.1]$; $fc = I_{Mu8.1}/I_{Ctr}$; fc: fractional current; $I_{Mu8.1}$: current in the presence of Mu8.1; $I_{Ctr}$: current in the absence of toxin. NB = no block; SEM = standard error of the mean; $n$ = number of experiments. Source data are provided in S4 Data.
(PDF)

**S1 File. Materials and methods for experimental work presented in supplementary information.**
(DOCX)

**S1 Data. Source data related to the identification of the con-ikot-ikot–like 4 clusters presented in Figs 1, S2-S4 and S14.**
(ZIP)

**S2 Data. Source data for SAXS analysis presented in Figs 2D and S6.**
(XLSX)

**S3 Data. Source data for effect of Mu8.1 on calcium influx in DRGs presented in Fig 6.**
(XLSX)

**S4 Data. Source data for electrophysiology experiments assessing Mu8.1 modulatory effect of Cav2.3 channel as well as Mu8.1 and SNX-482 effect on calcium influx in DRGs presented in Fig 7.**
(XLSX)

**S5 Data. Source data for intracellular $Ca^{2+}$ imaging experiments assessing the ability of Mu8.1 to block desensitization of GluA2 AMPA receptors shown in S8 Fig.**
(XLSX)

**S6 Data. Source data for assessing the current–voltage relationship of Cav2.3-mediated currents in the presence of Mu8.1 presented in S9 Fig.**
(XLSX)

**S7 Data. Source data for time course of Mu8.1 inhibition/recovery of Cav2.3-mediated current presented in S10 Fig.**
(XLSX)

**S8 Data. Source data for Mu8.1 binding in receptor and GPCRome screen presented in S11 Fig.**
(XLSX)

**S9 Data. Source data for Mu8.1's irreversible inhibition of $Ca^{2+}$ influx in neuronal subtypes presented in S12 Fig.**
(XLSX)

**S10 Data. Source data for experiments showing SNX-482 amplification of calcium signal in subpopulations of neuronal cells in DRGs presented in S13 Fig.**
(XLSX)

**S1 Raw Images. Original SDS-PAGE and Tricine gels as well as western blot images for data shown in Fig 2A and 2B.**
(TIF)

## Acknowledgments

We acknowledge the MAX IV Laboratory for time on Beamline Biomax under Proposal 20190334, and the University of Copenhagen, Small Angle X-ray facility, CPHSAXS (https://drug.ku.dk/core-facilities/cphsaxs/). We thank Dr. Uwe Müller for assistance during the data collection and Cecilie L. Søltoft for expert technical assistance. Radioligand binding and GPCR binding assays were generously provided by the National Institute of Mental Health's Psychoactive Drug Screening Program, Contract # HHSN-271-2018-00023-C (NIMH PDSP). The NIMH PDSP is Directed by Dr. Bryan L. Roth at the University of North Carolina at Chapel Hill and Project Officer Jamie Driscoll at NIMH, Bethesda, Maryland, USA.

## Author Contributions

**Conceptualization:** Celeste M. Hackney, Helena Safavi-Hemami, Lars Ellgaard.

**Data curation:** Paula Flórez Salcedo, Rocio K. Finol-Urdaneta.

**Formal analysis:** Celeste M. Hackney, Paula Flórez Salcedo, Emilie Mueller, Thomas Lund Koch, Pernille Sønderby Tuelung, Rocio K. Finol-Urdaneta, Jens Preben Morth.

**Funding acquisition:** Jeffrey R. McArthur, David J. Adams, Anders S. Kristensen, Baldomero Olivera, Helena Safavi-Hemami, Jens Preben Morth, Lars Ellgaard.

**Investigation:** Celeste M. Hackney, Paula Flórez Salcedo, Emilie Mueller, Thomas Lund Koch, Lau D. Kjelgaard, Maren Watkins, Linda G. Zachariassen, Pernille Sønderby Tuelung, Jeffrey R. McArthur, Rocio K. Finol-Urdaneta, Helena Safavi-Hemami, Jens Preben Morth.

**Methodology:** Celeste M. Hackney.

**Project administration:** Celeste M. Hackney, Lars Ellgaard.

**Supervision:** Anders S. Kristensen, Baldomero Olivera, Helena Safavi-Hemami, Jens Preben Morth, Lars Ellgaard.

**Visualization:** Celeste M. Hackney, Thomas Lund Koch, Linda G. Zachariassen, Pernille Sønderby Tuelung, Rocio K. Finol-Urdaneta.

**Writing – original draft:** Celeste M. Hackney, Paula Flórez Salcedo, Emilie Mueller, Thomas Lund Koch, Linda G. Zachariassen, Baldomero Olivera, Rocio K. Finol-Urdaneta, Helena Safavi-Hemami, Jens Preben Morth, Lars Ellgaard.

**Writing – review & editing:** Celeste M. Hackney, Paula Flórez Salcedo, Emilie Mueller, Thomas Lund Koch, Maren Watkins, Jeffrey R. McArthur, David J. Adams, Baldomero Olivera, Rocio K. Finol-Urdaneta, Helena Safavi-Hemami, Jens Preben Morth, Lars Ellgaard.

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
