## [Editor Report · Decision Letter 0]

25 Jan 2023

Dear Dr Ellgaard, 

Thank you for submitting your manuscript entitled "Identification of a sensory neuron Cav2.3 inhibitor within a new superfamily of macro-conotoxins" for consideration as a Research Article by PLOS Biology. Please accept my apologies for the delay in getting back to you as we consulted with an academic editor about your submission. 

Your manuscript has now been evaluated by the PLOS Biology editorial staff, as well as by an academic editor with relevant expertise, and I am writing to let you know that we would like to send your submission out for external peer review.

Once your full submission is complete, your paper will undergo a series of checks in preparation for peer review. After your manuscript has passed the checks it will be sent out for review. To provide the metadata for your submission, please Login to Editorial Manager (https://www.editorialmanager.com/pbiology) within two working days, i.e. by Jan 27 2023 11:59PM.

Kind regards,

Richard

Richard Hodge, PhD

Associate Editor, PLOS Biology

rhodge@plos.org

PLOS

---

## [Decision Letter · Decision Letter 1]

3 Mar 2023

Dear Dr Ellgaard,

Thank you for your patience while your manuscript "Identification of a sensory neuron Cav2.3 inhibitor within a new superfamily of macro-conotoxins" was peer-reviewed at PLOS Biology. Please accept my apologies for the delays that you have experienced during the peer review process. Your manuscript has now been evaluated by the PLOS Biology editors, an Academic Editor with relevant expertise, and by three independent reviewers. 

In light of the reviews, which you will find at the end of this email, we would like to invite you to revise the work to thoroughly address the reviewers' reports.

As you will see, the reviewers find your study interesting and well done, but ask that additional electrophysiological experiments are included to strengthen the claims regarding the selectively of the Mu8.1 conotoxin for Cav.2.3 channels, including the addition of a current-voltage curve for Cav2.3 and testing the effect of the toxin on Cav1 family members. In addition, they ask that the physiological experiments are strengthened to help to provide a mechanistic link for the observations, such as including an experimental comparison to what is observed upon SNX-482 injection. 

Given the extent of revision needed, we cannot make a decision about publication until we have seen the revised manuscript and your response to the reviewers' comments. Your revised manuscript is likely to be sent for further evaluation by all or a subset of the reviewers.

**IMPORTANT - SUBMITTING YOUR REVISION**

*Re-submission Checklist*

*Published Peer Review*

*PLOS Data Policy*

*Blot and Gel Data Policy*

Sincerely,

Richard

Richard Hodge, PhD

Associate Editor, PLOS Biology

rhodge@plos.org

REVIEWS:

Reviewer #1: This manuscript describes the isolation and structural characterization of a very large type of conotoxin. Most of the manuscript deals with the structural aspects which are valuable and novel. Where a little bit more work is needed is on the biophysics and physiology that is also described in the manuscript. 

Selectivity and electrophysiology: 

The authors did a nice job in looking at selectivity between Cav2.3 and other types of ion channels, including members of the Cav2 and Cav3 family. Based on this, I am quite convinced that this molecule is a blocker of this channel, albeit with relatively low 5 micromolar affinity. What is missing in this though is:

IV curve for Cav2.3, and testing whether there is activation gating inhibition 

time course of development and recovery from block

testing the effect of the toxin on a representative member of the Cav1 family

Physiology:

The physiological studies that are described are not contextualized properly vis a vis what is know with the role of R-type channels in mice. Cav2.3 null mice are not sleepy but they show reduced seizure activity in epilepsy models, so it is entirely unclear why the sedative effects were observed here. It is likely that another target is involved in these processes, and this needs to be discussed in detail. Along these lines, no proper discussion is provided concerning the zebrafish data, nor is it clear that R-type channels are responsible for the observed effects and the authors do not even try and provide a mechanistic link between R-type block and what is seen in these experiments. In fact, even the title of the figure legend is extremely vague, i.e. "the toxin induced a behavioral response". It seems to me that in both cases, an experimental comparison to what is observed upon injection of SNX-482 would be quite essential, and a much more detailed discussion of all of these matters. The n values in these experiments are also much too low (3 and 2). Along these lines, one of the most prominent roles of R-type channels is their contribution to nociceptive signaling and their role in seizure disorders, including kainate induced seizures. It seems to me that the authors could have tested the effect of the toxin in a mouse pain or seizure model. Alternatively, maybe the authors could simply leave out the in vivo experiments and focus in more detail on the target interaction with Cav2.3 as per my first point above. 

Supplementary Figure 11: No quantification or n values are provided for these data unlike in Figure S10. 

Reviewer #2: This manuscript reports the discovery and initial characterization of a novel class of conotoxins, Mu8.1. In contrast to most conotoxins, Mu8.1 is large (89 residues). The authors provide extensive biochemical, structural, and functional analysis. The structural data show that Mu8.1 has a saposin fold and has a reasonable propensity to form dimers. There are clear functional effects in both zebrafish and mouse models that provide evidence that the toxin is bioactive. Profiling shows that Mu8.1 has the highest potency against the R-type calcium channel CaV2.3 (a channel isoform that has rather poor pharmacology). The IC50 for this channel is not particularly high (~5 µM) and not terribly separated from its effects on other voltage-gated channels (~30 µM for a number of related CaVs, as well as a voltage-gated potassium channel). This 'preferential' activity that is mentioned in the abstract needs to be made a bit clearer as without a careful read of the manuscript, readers might get the impression that Mu8.1 is selective for CaV2.3. Nevertheless, overall the manuscript presents an important advance. There are a number of issues with the clarity of the presentation that should be addressed as 

Abstract last sentence. It is not clear what 'feasibility of large, disulfide rich venom component investigation' means. Maybe the authors are referring to the fact that they are able to make a bacterial system that can express the toxin? Although this is an important technical step, it probably also could be done by secreting the protein from insect or HEK cells. It seems like a strange ending sentence for the manuscript as I do not think this is the most important finding of the study.

Line 23 'asymmetric, heteromeric'. By definition, if a K channel is heteromeric it will be asymmetric.

Lines 130-135 The discussion on the similarity (or not) to the con-ikot-ikot toxins is confusing. First, it seems that the authors are trying to make the comparison based on the signal sequences, which is odd as what matters is whether the toxins themselves have similar folds in the part that is actually active. This is what the authors eventually get to in the MS, so what starts as a 'misannotation' is actually something that was correct. Since both Mu81. And the con-ikot-ikot toxins share the saponin fold, it would seem that this fact should be described in a less round-about way. The same fold appears to be used in different ways to engage different targets (glutamate receptors vs. CaV2.3 and relatives). The latter seems like a key point that is undersold here.

Line 404 Disulfide bonds are crosslinks. The are not metaphorical crosslinks and should not be in quotation marks.

Reviewer #3: The present manuscript reports interesting data concerning the identification of the sequence of an unusually large conotoxin, Mu8.1, that defines a new class of conotoxin that inhibits with a preferential selectivity the R-type (Cav2.3) calcium channel. Functional studies using defined classes of murine somatosensory dorsal root ganglion (DRG) neurons show that Mu8.1 effectively blocks the calcium influx across the neurons. When tested on a variety of voltage-gated calcium channels heterologously expressed in HEK 293 cells, Mu8.1 inhibited the Cav2.3) channel with an IC50 of about 5 microM, i.e, only 5 times lower than the other Cav channels (Cav2.1, Cav2.2 and Cav3), see Table S4. Although not impressively effective these new findings highlight the potential of Mu8.1 as a molecular tool to identify and study neuronal subclasses expressing Cav2.3. 

Electrophysiological experiments are on sound but the number "n" should be increases to 4-5 (see Table S4). I do not have major criticisms except suggesting the authors to smooth the emphasis on the potency of the new toxin as a Cav2.3 channel blocker. SNX-482 is far more potent than Mu8.1 and would be hardly replaced in pharmacological studies when assaying Cav channel availability in excitable cells. Nevertheless Mu8.1 represents a good new start concerning the underexplored group of macro-conotoxins that may lead to the discovery of new molecules with high affinity for Cav channels (highly required!).

---

## [Decision Letter · Decision Letter 2]

8 Jun 2023

Dear Dr Ellgaard,

Thank you for your patience while we considered your revised manuscript "Identification of a sensory neuron Cav2.3 inhibitor within a previously unrecognized superfamily of macro-conotoxins" for publication as a Research Article at PLOS Biology. This revised version of your manuscript has been evaluated by the PLOS Biology editors, the Academic Editor and Reviewer #1.

Based on the review from Reviewer #1 and our Academic Editor's assessment of your revision, I am pleased to say that we are likely to accept this manuscript for publication, provided you satisfactorily address the following data and other policy-related requests that I have provided below (A-D):

(A) We would like to suggest the following modification to the title:

"A previously unrecognized superfamily of macro-conotoxins includes an inhibitor of the sensory neuron calcium channel Cav2.3"

(B) Thank you for already providing the individual numerical values that underlie the summary data displayed in the figure panels (Source_Data). However, we note that the following figure panels are missing from the Source Data file and we would be grateful if you could include this data upon re-submission. It also seems that Figure S7 is mislabelled in the file (should be Figure S8?) Please ensure that all of you files are correctly labelled in the Source_Data file upon resubmission. 

Figure 2D, S10, S11A-C, S12A-B, S13A-B

(C) Please also ensure that each of the relevant figure legends in your manuscript include information on *WHERE THE UNDERLYING DATA CAN BE FOUND*, and ensure your supplemental data file/s has a legend.

(D) Please ensure that your Data Statement in the submission system accurately describes where your data can be found and is in final format, as it will be published as written there. 

We expect to receive your revised manuscript within two weeks. 

*Published Peer Review History*

*Press*

Kind regards,

Richard

Richard Hodge, PhD

rhodge@plos.org

Reviewer remarks:

Reviewer #1: OK with me as is

---

## [Editor Report · Decision Letter 3]

27 Jun 2023

Dear Lars,

On behalf of my colleagues and the Academic Editor, Thomas Sudhof, I am pleased to say that we can accept your manuscript for publication, provided you address any remaining formatting and reporting issues. These will be detailed in an email you should receive within 2-3 business days from our colleagues in the journal operations team; no action is required from you until then. Please note that we will not be able to formally accept your manuscript and schedule it for publication until you have completed any requested changes.

PRESS

Best wishes, 

Richard

Richard Hodge, PhD

rhodge@plos.org

PLOS
